# A Review of RFID Sensors, the New Frontier of Internet of Things

**DOI:** 10.3390/s21093138

**Published:** 2021-04-30

**Authors:** Filippo Costa, Simone Genovesi, Michele Borgese, Andrea Michel, Francesco Alessio Dicandia, Giuliano Manara

**Affiliations:** 1Dipartimento di Ingegneria dell’Informazione, Università di Pisa, 56126 Pisa, Italy; simone.genovesi@unipi.it (S.G.); andrea.michel@unipi.it (A.M.); giuliano.manara@unipi.it (G.M.); 2Research and Development Department, Siae Microelettronica, 20093 Cologno Monzese, Italy; michele.borgese@for.unipi.it; 3IDS Ingegneria dei Sistemi SpA, 56121 Pisa, Italy; f.dicandia@idscorporation.com

**Keywords:** RFID, sensors, RFID sensors, RFID antennas, RFID readers, chipless RFID, wireless sensors, wireless sensors networks, internet of things, metamaterials, metasurfaces, printed electronics

## Abstract

A review of technological solutions for RFID sensing and their current or envisioned applications is presented. The fundamentals of the wireless sensing technology are summarized in the first part of the work, and the benefits of adopting RFID sensors for replacing standard sensor-equipped Wi-Fi nodes are discussed. Emphasis is put on the absence of batteries and the lower cost of RFID sensors with respect to other sensor solutions available on the market. RFID sensors are critically compared by separating them into chipped and chipless configurations. Both categories are further analyzed with reference to their working mechanism (electronic, electromagnetic, and acoustic). RFID sensing through chip-equipped tags is now a mature technological solution, which is continuously increasing its presence on the market and in several applicative scenarios. On the other hand, chipless RFID sensing represents a relatively new concept, which could become a disruptive solution in the market, but further research in this field is necessary for customizing its employment in specific scenarios. The benefits and limitations of several tag configurations are shown and discussed. A summary of the most suitable applicative scenarios for RFID sensors are finally illustrated. Finally, a look at some sensing solutions available on the market are described and compared.

## 1. Introduction

Radio frequency identification (RFID) is a low-cost wireless technology that makes possible the connection of billions of things, enabling consumers and businesses to engage, identify, locate, transact, and authenticate products [1]. The general RFID market has seen a considerable growth over the past few years in terms of the number of RFID tags sold. The exploration of allied technologies such as RFID sensors has been enabled thanks to new chipsets, both within the High Frequency (HF) band (NFC—near field communication) at 13.56 MHz, and within the Ultra-High Frequency (UHF) (RAIN—radio frequency identification) frequency band around 866 MHz (ETSI - European Telecommunications Standards Institute) or 910 MHz (FCC - Federal Communications Commission), which are dedicated to supporting several sensor platforms [2]. Even though battery-powered sensors have an obvious advantage for data transmission over large distances, at the same time, a battery increases the system’s complexity and maintenance issues, reduces system life, and limits the temperature range of sensor applications. Small form factors of batteries, such as thin-film or other types of micro-batteries, are available on the market, but they need replacement every few days [3]. RFID sensors can be fully passive, or battery powered; in the latter case, they do not need a frequent change of the batteries as in the case of traditional wireless sensor nodes. In case of battery assisted RFID sensors or battery-assisted passive mode, a simple circuit is built around the memory chip, thus enabling the chip to switch to a local energy-assisted mode only when it senses a certain stimulus [4]. Instead, chip-based passive sensors acquire the power required for activation from the reader through wireless power transfer. 

Different RFID sensors are currently proposed in terms of architecture, complexity, and system requirements. A chip-based design, where the sensor is integrated inside the chip, provides a reliable configuration, since the sensing and communication functions are separated. Since embedding the sensor increases both tag size and cost, an alternative solution is the functional integration of the antenna and the sensor component. The challenge is then to transform the RFID tag antenna into a sensor. In antenna-based RFID sensors, the response is more dependent on the environment. Today’s generation of passive tags has the ability to sense several environmental parameters such as light, humidity, and temperature, becoming a key technology for “object-based” services. The role of RFID systems can be extended, and even involve ubiquitous computing by moving from simple passive tags to smart tags that can perform different functions, thanks to integrated sensors and a microcontroller unit (MCU). Research is also active on the so-called chipless RF sensors that do not employ Integrated Circuits (ICs) and may offer the benefits of a longer life and lower cost. Chipless RFID, also known as passive RFID sensors, are compatible with planar technology, allowing them to be produced by roll-to-roll processing. 

RFID sensors are a new paradigm for the internet of things (IoT). They have a limited cost and negligible maintenance, which make them appealing for numerous applicative scenarios such as manufacturing, logistics, healthcare, agriculture, and food. They have attracted numerous research efforts due to their innovative potential in various application fields [5]. In order to also address the relevance of the research topic at an industry level, we report some figures obtained from the database Scopus. In particular, the number of patents published every year according to the keywords ‘RFID’ and ‘RFID sensor’ are reported in Figure 1a. The patents classified by registration offices are reported Figure 1b to highlight the most relevant countries active on this technology. We observe that a very high number of patents, i.e., more than 129,000, have been registered with the United States patent office. Surprisingly, the number of papers published in journals and conferences every year is one order of magnitude lower than the number of registered patents. 

## 2. From Wi-Fi Nodes to RFID Sensors

The typical hardware platform of a wireless sensor node consists of a sensor, a microcontroller, a radio frequency transceiver, and a power source. Each node is equipped with a physical sensor for revealing parameters of interest, such as light, temperature, sound, pressure, or other physical phenomena. The architecture of a wireless sensor node is shown in Figure 2. Power consumption, chip footprint, and computational power, as well as on-chip memory are very important features for a microcontroller. To this aim, choosing a low-power-consumption transceiver that connects the node to the network is crucial for saving power. In fact, current consumption of a transceiver takes up most of the power consumption budget. Transceivers are available from various manufacturers, such as Infineon Technologies AG (Neubiberg, Germany) [6], Analog Devices Inc. (Norwood, MA, USA) [7], and others. A crucial characteristic of a sensor node is the power source and battery management, especially in wireless sensor networks (WSNs), where the battery is irreplaceable. The node typically also includes a circuit that adjusts the RF front-end to compensate for changes of the RF front-end caused by the sensing element [8].

Although battery technology is mature, the mean time to replacement is only one year, or two, even, for relatively large batteries [9]. This means that for a sensor network with a considerable number of devices installed, several batteries need to be replaced every few days, which is unfeasible for many applications. Finally, systems based on battery-powered communication still cost more than USD 5 a unit, with no clear commercial or technical solution to achieve a cost lower than one dollar [9]. A possible way to overcome this problem is represented by ambient-power scavenging, which can, in principle, supply power indefinitely. In this context, an augmented version of RFID technology with sensing capabilities can represent the turn-around point for the massive development of the internet of things (IoT). 

RFID sensors can be of different types, and are based on many different approaches. Some of the sensors are already available on the market, while other are still in development and are not yet mature enough for market applications. A classification of RFID sensors covering both chip-equipped tags and chipless tags is shown in Figure 3. The former set includes configurations in which the sensor is integrated directly into the tag (electronic sensors), whereas the latter rely on the modification of the tag response due to a detuning of the tag antenna for sensing (electromagnetic sensors). On the other hand, the chipless counterpart can exploit the properties of piezoelectric materials (acoustic sensors), the changes in the electromagnetic response of the tag (electromagnetic sensors), or thin-film transistors (electronic sensors) for sensing purposes.

Finally, it is important to point out that each one of the mentioned solutions for sensing possesses pros and cons. Consequently, the user has to decide the best trade-off, depending on the goal they want to achieve. A summary of the characteristics of various kinds of sensors is illustrated in Table 1, where it is emphasized, once again, that higher performance comes at the cost of a higher price.

## 3. Chip Based RFID Sensors

There exist two implementations of a sensor employing a chip, namely the electronic RFID sensor and the electromagnetic one. A sketch representing the two topologies is shown in Figure 4. Both configurations contain the same functional units, but with a fundamental difference in the sensing part. In the electronic configuration, the sensing unit interacts with the IC, whereas in the electromagnetic configuration, the sensing operation is based on the change in the tag frequency response. In both cases, the primary constraint that determines the maximum reading distance of the sensor tag is represented by the value of the voltage that is available at the chip [10].

### 3.1. Electronic RFID Sensors Tags

Electronic sensor tags have the peculiarity of separating the sensing functions from the communication functions [11]. The transmitted information is digitally encoded, and thus the sensed information is essentially insensible to the environment. In particular, some bits of the ID code can be used to transmit the value of the sensed parameter. The sensor can be integrated inside the chip, or interfaced though an external microcontroller, thus obtaining an augmented tag architecture. In both cases, the energy efficiency of the tag is a key issue for achieving an acceptable reading range. When the reading range is not enough, a battery is included within the tag in the so-called battery assisted tags. In these cases, the tag acts like a transponder, and thus it sends the information to the reader only if interrogated. When the tag has to continuously broadcast the sensor information, an active configuration has to be used.

#### 3.1.1. Active Sensor Tags 

In active RFID tags equipped with sensing capabilities, a sensor is integrated into the tag, and the tag IC communicates with the sensor in order to get the information on the monitored quantity, and thus to include it into the backscattered bit sequence. The sensing activity is therefore performed in DC with a dedicated electronic component. The integration of the sensors within the chip circuit requires the design of dedicated chips with specific sensing features. Being equipped with a battery, active tags provide a much longer read range than passive tags. Active RFID systems generally work at 433 MHz or 2.45 GHz [12], but 433 MHz is usually preferred by companies because of the longer wavelengths, which result in being more suitable for materials like metal and water. Typically, active RFID tags are powered by a battery that will last between 3–5 years. Active RFID tags can be employed both as transponders and beacons. Beacons have a read range of hundreds of meters but, in order to limit battery consumption, their transmitting power can be set to reach a read range of around 100 m. Active RFID tags can cost from $20 to $100+ depending on the tag specifications and on its capability to withstand harsh conditions [12]. Active RFID tags can be equipped with motion and temperature sensors. Recently, novel configurations of 2.4 GHz and 5.7 GHz active tags are available, which can be configured to activate when moved, record temperature, humidity or other environment parameters, and transmit signals periodically. The most typical sensing applications of active tags are as follows [13]:-Real-time locating systems (RTLS): it is possible to determine the location of active tags with correct antenna placement.-Patient tracking: the movements of patients can be monitored within a certain area with a wrist band tag.-Temperature sensing: active tags can be used to sense temperature variation in time with alarms set if a certain threshold is overtaken.-Motion sensing: the tag can be set up to transmit a signal when moved or a signal until movement stops. This is used to control the movement of high value assets and for avoiding intrusions.

#### 3.1.2. Battery Assisted Sensor Tags

The battery assisted passive (BAP) class is comprised of RFID tags with an embedded battery. When the reader interrogates the BAP tag, the embedded battery is turned on, as well as the RFID tag IC and other sensors or actuators within the tag [14]. BAP RFID tags’ read range is clearly greater than passive ones, but is shorter than active tags. The operative life of these tags is limited by the battery, which is usually not replaceable. As an example, the Monza X-2K RFID chip manufactured by Impinj Inc. (Seattle, WA, USA) [15] provides a performance boost via a ‘battery-assisted passive mode’ [4], when both read sensitivity (−17 dBm) and write sensitivity (−12 dBm) of the chip increase up to −24 dBm when a DC voltage is provided [4]. Figure 5 illustrates a scheme of RFID tags relying on battery-assisted sensors. To limit the use of the battery, thus increasing the battery-life, energy harvesting solutions have been proposed and integrated into both active and battery-assisted RFID tags [16]. Technologies relying on thermoelectric effects, photovoltaic effects, or piezoelectric effects have been adopted for harvesting energy from the surrounding environment. Conversion of biomechanical energy into electricity has also been proposed in [17].

#### 3.1.3. Battery-Less Sensors (Passive)

Solutions that provide sensors integrated within the chip have been recently proposed [10,18]. A sensor is embedded and communicates with the IC to deliver information on the monitored quantity, which has to be encoded into the backscattered signal. Electronic RFID sensors can exploit the HF or UHF frequency range. An example of a reader for HF sensors is represented by smartphones exploiting the near field communication (NFC) paradigm. For UHF tags, which cover a longer communication range, the amount of power rectified by the tag must be sufficient to power up both the chip and the embedded sensor. In this case, the value of the voltage that is available at the chip [10] is pivotal for the reading range extension [19]. Obviously, the sensor must be able to be miniaturized in order to be embedded into the chip, and few examples are currently available in the literature dealing with temperature, light, and pressure monitoring. However, the passive RFID tag chip is so sensitive to power consumption that it is difficult to embed sensors and an analogue-to-digital converter (ADC) maintaining a reasonable operating range [20]. Even an ADC with a power consumption of several µW [21] will reduce the operating distance of the tag dramatically.

A new augmented RFID tag, the WISP (wireless identification and sensing platform), has been developed by the Intel Research Centre [22]. WISP operates as a standard tag, but offers the possibility to be connected to an external sensor by using a programmable microcontroller unit. However, the flexibility offered by the WISP solution is paid for with an increased price. Battery-less configurations have recently been developed in which the microcontroller requires a few milliwatts [4]. Moreover, they offer the possibility of gathering the energy required to drive the data acquisition from the interrogation signal, as in the case of conventional passive tags. Another option is represented by piezoelectric energy scavengers transforming micro-oscillations into electrical energy, which are the most powerful and versatile ones, although the power requirements may reduce the real data rate. 

### 3.2. Electromagnetic RFID Sensors Tags

The effective permittivity of an object can be inferred by using any conventional RFID tag attached to it. In fact, quantities such as the input impedance or gain of the RFID tag, as in the case of any antenna, are altered by the surrounding environment. The RFID backscattered signal is affected by any change of the tagged object, since this is shifted into a correspondent change in the tag’s antenna parameters. Therefore, the antenna itself becomes the sensor, even though it is completely sensor-less [23].

Electromagnetic tags rely on the modification of the tag response due to an occasional modification of the antenna. The antenna behavior can be modified essentially for two reasons: a change in the electrical conductivity of the antenna, or a part of it; or because of a change of the dielectric permittivity of the medium surrounding the antenna, or a part of it. In the first case, the sensor is classified as resistive, and in the second case, the sensor can be classified as capacitive. Notably, in both cases, the transduction mechanism is influencing radio frequency waves and thus the sensors are referred to as electromagnetic. 

A possible layout of the RFID sensor is shown Figure 6a, where a dipole antenna is connected in series with a sensing unit and the RFID chip [24]. A simplified equivalent circuit model of an electromagnetic RFID-based sensor tag is shown in Figure 6b. The sensing material, e.g., carbon nanotubes, Kapton, or other substances, which change their properties as a function of an external stimulus, are placed in the sensing unit. The variations of the sensing unit properties impact on the matching between the antenna and the chip impedance, and thus they induce a change of the reflected power. Depending on the value of the sensed parameter, the impedance mismatch between the antenna port and the microchip is enough to enable (or not) the microchip to harvest the necessary power and respond to the reader with its own digital identification code (IDn) [23]. 

The two main mechanisms for detecting a variation in the sensor response are a frequency shift (Δf) of a resonant peak or a magnitude change (Δα) in the reflection coefficient, as depicted in Figure 6c,d. The frequency shift method is not compatible with RFID chip-enabled technology, since the resonant frequency shift of the antenna leads the information content far away from the operating frequency (868 MHz or 915 MHz). Therefore, magnitude change configuration is the most popular approach for RFID-based sensors. 

An alternative sensor topology consists of equipping the tag with a real sensor (motion, temperature, pressure, or other), which could be either connected in a region of the tag’s antenna or distributed over the antenna surface as a paint. The variation in the impedance loading caused by the change of the environment will produce a change of the tag’s radar cross-section and, hence, a backscattered power modulation, as in the case of sensor-less tags [23].

#### 3.2.1. Self-Tuned Chips

In the case of electromagnetic RFID sensor tags, the antenna accomplishes two different functions, namely the communication and the sensing tasks. However, they cannot be pursued independently or maximized at the same time, since the sensing is performed at the expense of the communication function. In fact, the tag must suffer a certain level of mismatch in order to indirectly provide information about the variation of the measurements. The reason for this is that the sensing information is strictly related to the detuning level of the RFID tag, and hence the communication and sensing capabilities have conflicting requirements. A recent solution to this problem has been offered by self-tuning microchips [25,26,27] that can cope with the undesired tag detuning without harming the sensing function [28]. The equivalent circuit of an RFID tag with self-tuning capability is illustrated in Figure 7a. The working principle can be explained by considering that the RFID tag is placed on an item that drastically alters its antenna impedance with respect to the free space case. This means that the original antenna is detuned, and the impedance Z_A_ = R_A_ + jX_A_, which was designed to properly match the impedance of the chip, Z_C_, may be far from the optimal condition Z_C_ = (Z_A_)*, and therefore the communication may be severely compromised.

The integrated circuit of a self-tuning RFID tag is able to guarantee a proper level of matching thanks to its ability to autonomously adjust the chip’s impedance to the current antenna impedance. This goal is achieved for the chip via a set of capacitors that can be selectively activated. More specifically, the chip reactance spans from a minimum value of C_min_, where all the other N selectable capacitors, C_S_, are switched out, up to the maximum value of C_max_ = C_min_ + N C_S_ that is attained when all the capacitors are activated. The configuration status of all the capacitors in the switching network is automatically selected to assure the maximum power transfer from the antenna to the microchip. This means that the microchip neutralizes the undesired changes of the RFID antenna impedance with the aim of reaching the closest condition to perfect matching.

The adaptive matching feature exhibited by this kind of RFID tag can also be advantageously exploited for sensing purposes. In fact, the self-tunable chip sends to the RFID reader the information written in its memory, as well as the state of the selectable capacitors. This string of bits is related to the detuning undergone by the RFID tag, and indirectly provides information on the environmental change in which the tag is operating. It is then possible to infer the value of a physical parameter that affects the tag from the state of the automatic tuning network. Figure 7b illustrates an applicative scenario in which an RFID tag is applied to a container in which the level of water (h_WATER_) can change over time. Every time the tag is interrogated, it also provides the status of the internal matching network that allows it to be matched and running in the complex operational environment. The encoded sequence of the capacitors selected in the tunable matching network at time t_1_ is embedded in the backscattered signal. When the water level is changed at time t_2_, the tunable matching network is set to the new status of the capacitors, since the effect of the environment has changed. The change of the matching network settings that has been encoded in the backscattered signal allows for indirectly estimating the level of the water. Obviously, the limited number of capacitors in the switching network, and also their range of values, have an impact on the resolution of the monitored physical parameter [29].

#### 3.2.2. Harmonic Tags

An alternative implementation of RFID sensors is based on the so-called harmonic or secondary radar [30]. In this configuration, the tag receives the interrogation at a certain frequency, and it scatters back at a harmonic frequency. A typical implementation of harmonic tags employs the second harmonic frequency (2f_0_). 

The reason for using harmonic tag configuration relies on the possibility of obtaining a better immunity of the tag response in the presence of clutter. Indeed, it can be reasonably assumed that most of the objects surrounding that tag do not possess nonlinear properties, which may cause the reflection of harmonic frequencies. 

An harmonic transponder is usually synthesized through a Schottky diode and an antenna [30,31,32]. The Schottky diode should be able to perform the frequency multiplication as efficiently as possible to 2f_0_. The antenna allows for radiating the upconverted frequency signal. One issue is that the antenna matching should be guaranteed both at the fundamental and the second harmonic frequencies. This aspect makes the transponder design more complex if compared to standard RFID tags. An additional challenge is that the tags need to comply with existing frequency regulations.

## 4. Chipless Sensors

Chipless RFID sensor tags, like the electromagnetic RFID sensors, exploit the changes in antenna behavior that is dependent on the change in the physical environmental parameter that has to be measured [33]. However, differently from classical electromagnetic RFID sensors, the tag does not include a chip. It is basically a simple passive antenna or resonator. An electromagnetic wave impinging onto the chipless tag is mostly backscattered at the resonance frequency of the resonator, which acts as a spatial filter. Therefore, by observing the spectral response of the backscattered signal, a peak can be observed at the resonance frequency of the resonator, or resonant frequencies, in the case of multiple resonators. This technology appears to be promising for designing low-cost, green, and printable sensors [34]. The sensing capabilities are obtained through a frequency shift of the resonant peaks in the backscattered response generated by the change in external parameters surrounding the tag. The absence of a chip and a battery gives the opportunity to significantly decrease the cost of the sensor and to achieve a theoretically infinite lifetime. Given the absence of any electronic circuit, chipless RFID sensors are potentially suitable for harsh environments [33,35,36,37]. Clearly, one of the main limitations is that the reliable reading of the sensor can be guaranteed only under specific conditions. Chipless RFIDs rely on passive transduction mechanisms such as capacitance or surface resistance change, but even a mechanical perturbation of an RF resonatorallows for realizing cheap devices [36,38,39]. Printable electronic circuits can also be included in the chipless category. A summary of three main categories of chipless sensors is shown in Table 2. The accuracy of chipless tags is expected to be more limited with respect to chip-equipped sensors, but their study is justified by their extreme fabrication simplicity, which would allow for integrating these sensors inside conventional packaging at a minimal cost. Some examples of chipless RFID sensors that are commercially available are summarized in Table 3. As it is apparent, the employed technologies are quite broad-spectrum, spanning from the exploitation of the Barkhausen effect to radar imaging.

### 4.1. Electromagnetic Chipless RFID Sensors

Two general types of chipless RFID tags can be identified: time domain (TD)-based and spectral (frequency) signature-based (frequency domain, FD). Time domain chipless sensors are comprised of a wideband antenna and a delay line. The tag receives and retransmits a modified version of the pulse transmitted by the reader. The time domain radar cross section (RCS) of the tag is comprised of a structural and an antenna component. The delay line is used to separate the antenna RCS peak from the structural one, which is the most intense. Depending on the load of the delay line, the antenna response of the tag can be modulated or delayed. Both these mechanisms can be used to design a chipless sensor. Amplitude modulation can be obtained by ending the delay line with a material able to change its surface resistance as a function of an external phenomenon. The sensed information can be also included in the time shift from the antenna peak [40]. Time domain-based chipless RFID tags manufactured with printed circuit board technology are large and are unable to encode a high number of bits [41].

**Table 3 sensors-21-03138-t003:** Chipless RFID sensor tags that are commercially available. From [42].

Company	Tecnology
HID	Barkhausen magnetic access card
AstraZeneca	Acoustomagnetic Syringe tag
RF saw, Microdesign	SAW
Menippos, M-real, Acreo	Conductive stripes
Inksure	Radar Array
Myyake, Navitas, RF code	LC array
Flying null, Confirm technologies	Electromagnetic

The second class of chipless tags encodes data into the spectrum using resonant structures [43,44,45,46,47]. Two configurations have emerged for FD chipless tags. The former is based on two orthogonally polarized antennas with a series of resonators in between [46], whereas the latter employs several resonators of different size [47]. In FD chipless sensors, the idea is to add a sensor function to the tag. This goal can be achieved by exploiting the permittivity variation of a chemical interactive material (CIM) that is placed on the tag antenna. The CIM can cause a frequency shift in the backscattered signal or a change in the amplitude response, as illustrated in Figure 6. Different environmental parameters can be monitored by exploiting the interaction of the CIM with resonators. Temperature, pressure, humidity, and gas sensors have been proposed in [48,49,50,51]. Inkjet printing can be also efficiently exploited for the fabrication of chipless sensors [52,53,54]. Several polymers and nanomaterials have been used to synthesize sensors because of their ability to change their properties as a function of environmental parameters such as humidity, temperature, and gasses, or due to mechanical stresses and pressure. These materials are named CIM as they are able to operate a transduction from a physical quantity to an electrically measured one. In Table 4, some materials reported in the recent literature are categorized with their functionalization property. The table is based on [55,56]. A simplified scheme of the sensing mechanism based on chipless tags is shown in Figure 8.

The use of a small tag provides a weak backscattered signal, which can hardly be distinguished from the background response. Beyond the specific topology of the tag, the main limitation of chipless technology is that the tag detection requires a calibration procedure often based on background subtraction. For this reason, some strategies to increase the immunity of the tag response with respect to the environment have been investigated. Several solutions have been proposed in order to overcome this fundamental problem, such as adopting encoding/decoding schemes based on cross-polarization [57,58], circular polarization [59], differential polarization encoding [60], or synthetic aperture radar (SAR) based approaches [44]. To further improve the system reliability, the above strategies can be jointly adopted with time domain gating. Time domain gating allows for filtering out some undesired contributions, such as antenna coupling and multipath. However, the presence of large objects behind the tag is difficult to remove with time domain gating, since the time window should fulfill a couple of opposite requirements: the time window should be long enough to include all the time domain responses of the resonant tag, which is inversely related to the frequency bandwidth of the peak, but, at the same time, it should be truncated to remove the contribution of the objects close to the tag [60]. If the object is too close to the tag, the time gating may be not sufficient to completely remove the undesired contributions. The possibility of correctly reading the information encoded within the tag (detection probability) is strictly related to the RCS of the tag. The RCS of the tag is proportional to the square of the footprint of the label if all the particles radiate in phase [61]. In the case of periodic tags, the average value of RCS can be controlled by increasing or decreasing the number of unit cells, and thus the footprint of the tag [61]. Finally, it has to be mentioned that readers for chipless RFID require complex and specialized architectures based on their mode of operation, and competitive commercial products are currently available. Examples of reader prototypes include frequency-modulated continuous wave (FMCW) or ultra-wideband (UWB) impulse radar-based devices [62,63].

### 4.2. Acoustic Chipless RFID Sensors: SAW Tags

A surface acoustic wave (SAW) tag [64] is comprised of a piezoelectric substrate, an inter-digital transducer (IDT), some metal reflectors, and antennas. When the RF reader interrogates the tag, the tag antenna collects the EM probing wave. The IDT then converts the electronic energy into mechanical energy through the inverse piezoelectric effect. The SAW undergoes a series of reflections that operate the encoding of the signal, and then these acoustic waves are reconverted by the IDT into an EM wave by piezoelectric effect. This signal is finally transmitted back to the reader by the antenna tag. The RF operation frequency is limited by the substrate size and by the photolithographic process. SAW devices are manufactured in the frequency range between 30 MHz and approximately 3 GHz.

SAW delay lines are based on the SAW propagation time delay, calculated as the ratio of acoustical length and SAW velocity (*L*/*v_SAW_*). In known sensor applications, *L* and *v_SAW_*, respectively, vary as a transducer effect determined by a temperature change, mechanical stress, and strain, and because of a mass loading from a thin surface layer. SAW sensors can measure physical, chemical, and biological parameters. As an example, a pressure sensor is reported in [65], where the monitored parameter is converted into a change in the sensor’s surface acoustic wave’s velocity. As shown in [66], by bending, stretching, and compressing the SAW substrate, sensors for torque, force, displacement, vibration, and acceleration can be synthesized [67]. The first SAW sensors were for temperature monitoring, and sensitive materials such as quartz (SiO2) were employed. In multi-sensor systems, different sensors are typically distinguished through frequency division multiplexing. However, space division multiple access (SDMA) and time division multiple access (TDMA) can also be employed [64]. Orthogonal frequency coding (OFC) has also been proposed for encoding the SAW sensor in multisensory environments [68]. 

### 4.3. Electronic Chipless RFID Sensors: Thin Film Transistors (TFT)

An attractive solution for designing chipless RFIDs would be to manufacture the antenna and electronics on the same substrate. Organic electronics could represent the solution that enables the production of complete ultra-low-cost RFID tags. Thin film transistor circuits (TFTCs) can be also employed in designing TD tags [69]. However, the current development stage of printed RFID tags is still not sufficient to enable real-world applications [70]. Despite the availability of printed organic transistors with carrier mobilities over 1 cm^2^ V^−1^ s^−1^ and switching speed in the MHz range [71], silicon electronics strongly outperform printed electronics in terms of carrier mobilities and switching speeds, as shown in Table 5. However, printed electronics still provide interesting characteristics related to manufacturing on soft substrates in large areas. Currently, transistor architectures have not been optimized for ICs because of the limited number of metal layers available [72].

TFT-based RFID tags have been developed by companies or by research groups. Some developed tags can communicate only with specially designed RFID readers with custom protocols (like 8, 12, or 16-bit tags), whereas other designs are capable of communicating with commercial NFC readers, which, nowadays, are also integrated into cellphones. Clearly, if simplified protocols that take into account the technology limitations of TFTs are employed, the system requirements for the chip turn out to be less stringent. The three main challenges for designing TFT-based NFC tags are the data rates of 106 kbit s^−1^, a 128-bit memory read-out, and a limited incident power at the tag, in the range of ~10 mW, from the smartphone [72].

Companies are putting great effort into finding a viable manufacturing process. PolyIC is the first company that demonstrated RFID tags produced by roll-to-roll printing. EVONIK develops unique oxide semiconductor materials, whereas IMEC and TNO recently made the OLAE chip with the largest memory (128 bits) and a rectifier working at ultrahigh frequencies. Other companies such as Dai Nippon Printing and OrganicID are working on printed electronics for realizing RFID. OrganicID’s proprietary technology, acquired by Weyerhauser, is supported by several U.S. and foreign patents and patent applications. This technology is used to generate a low-cost 13.56 MHz RFID tag for use in tracking items through manufacturing and shipping. However, the development of the technology is still in progress [74]. Despite significant efforts, printed circuits are still not on par with silicon ICs in terms of stability and performance. Hence, silicon ICs continue to dominate flexible hybrid electronics (FHE) implementations. 

## 5. Applications

Radio frequency identification has gained wide public interest, since RFID can be a valid alternative to traditional barcode technology and provides additional features with respect to other alternatives. Printed bar codes are typically read by a laser-based optical scanner that requires a direct line-of-sight to detect and extract information. On the contrary, RFID can be read even when the tag is concealed for either aesthetic or security reasons. Moreover, the low cost could boost the use of RFID tags as pervasive environmental sensors on an unprecedented scale [75]. RFID sensing applications are comprised of monitoring physical parameters, automatic product tamper detection, harmful agent detection, and noninvasive monitoring. Some applications require reading passive tags from a distance of a few centimeters, while others need to read active tags at several hundred meters. Every application poses different constraints and requirements, so a feasibility field study must be performed in the operative environment to choose the best frequency and tags. An overview of the most relevant fields of applications for RFID sensors is summarized in Figure 9.

### 5.1. Healthcare

The healthcare industry plays a pivotal role in many economies. For example, in the United States (U.S.), the budget is in the order of several trillions of dollars [76]. Analysts forecast that the number of older people in the U.S. will increase by 135% between 2000 and 2050, and that the “population aged 85 and over—probably the group needing health and long-term care services more than any other—should increase by 350%” [77]. This implies a significant rise in the budget that states should allocate for healthcare. A possible mitigation effect of this trend can be provided by the massive employment of RFID technology, as it allows for preventing errors, saving costs, increasing security, and providing improved quality of life for patients.

Cheap sensors (wearable, implanted, and environmental), integrated into the paradigm of the internet of things (IoT), have the potential to make real a personalized smart-health system, where the natural habitat of the person, their body, and the internet are collaborating to manage and increase overall medical knowledge. For example, by displacing wireless sensors inside the home, on clothes, and on personal items, it becomes possible to monitor the macroscopic behavior of the person, as well as to compile statistics, to identify precursors of dangerous behavioral anomalies, and finally to activate alarms or prompt for remote actions by appropriate assistance procedures. RFID systems may represent a strategic enabling component thanks to the energy autonomy of battery-less tags and their low cost, which make them compatible with extensive distribution and even disposable applications [78].

Assuming a large diffusion of cheap passive UHF RFID tags inside the environment, it is possible to infer information about human activity. For example, human body movements in close proximity to the tags may introduce scattering and shadowing effects, thus altering the communication link between a fixed reader and the tags [79]. Such changes in the signals received by a combination of wearable tags and ambient tags can be then used to monitor human activity. For example, the state of children and disabled and elderly people in domestic and hospital environments during the night can be monitored by installing a UHF RFID reader that continuously illuminates the bed and detects the presence of a patient in the bed, their body movements, accidental falls, or suspicious long periods of inactivity (which might be due to fainting, unconsciousness, or even death), as well as interactions with objects nearby (e.g., medicines or glasses). By also employing temperature or humidity RFID sensors, even fever evolution and urine loss could be taken under control. 

Printed sensors utilize printed materials to transduce physical quantities such as temperature, light intensity, sound, force, or chemical reaction to electrical signals. In a printer thermistor, for example, the temperature variation leads to a change in the resistance of the printed active material. For these reasons, in the last decades a large number of research studies have been focused on the design of printed sensors for wearable health monitoring, which may allow for a continuous measurement of all vital signs such as blood pressure, pulse oxygenation, body temperature, and heart and respiration rates. The system is also suitable for providing reports and aggregated statistics, useful for the formulation of diagnosis and for the follow-up of therapies [78]. An example of monitoring system is shown in Figure 10a.

Moreover, by monitoring the fluctuations of backscattered power from sensor-less passive tags placed over the body it is possible to collect information about the subject, such as standing or moving conditions, and, eventually, to estimate the frequency of periodic movements [80,81]. A measurement setup for gesture recognition based on multiple RFIDs is shown in Figure 10b.

### 5.2. Food

Intelligent labeling of food products to indicate and report their freshness and other conditions is one important possible application for RFID sensors. Indeed, the market demands new sensors for food quality and safety with battery-free operation and minimal sensor cost [48,82]. In these sensors, the electric field generated in the RFID sensor antenna is affected by the ambient environment, providing the opportunity for sensing. This environment may be in the form of a food sample within the electric field of the sensing region, or a sensing film deposited onto the sensor antenna. Examples of applications include the monitoring of milk freshness, fish freshness, and bacterial growth in a solution. Unlike other food freshness monitoring approaches that require a thin film battery for the operation of an RFID sensor and the fabrication of custom-made sensors, the passive RFID sensing approach combines the advantages of both battery-free and cost-effective sensor design, and offers response selectivity. For food spoilage or quality monitoring applications, most HF passive RFID sensors rely on measuring volatiles or analytes in food packaging, whereas LC resonator sensors are based on measuring changes in dielectric permittivity [83,84].

As food ripens and spoils, its chemical composition changes, resulting in changes in its dielectric properties. This changes the coupled capacitance, and, in turn, the resonant frequency of the sensor. These sensors are either attached to food packaging or placed on the surface of the food itself. The approach has been demonstrated using conformal adhesive LC resonators attached to the surface of a banana skin or cheese [85], as a 3D printed LC resonator integrated in a milk package cap [30], and as a planar LC resonator attached to the surface of a milk package (Figure 11). The depth of interaction between the food and the sensor (the penetration depth) depends on the operating frequency of the sensor, electric conductivity, and the dielectric polarization loss of the food [48].

### 5.3. Agriculture

The development of RFID systems for smart agriculture applications has attracted considerable research efforts in recent years [87]. The precision agriculture system, through the exploitation of RFID technology, allows farmers to maximize the yield by increasing efficiencies, productivity, and profitability, while minimizing unintended impacts on wildlife and the environment [88]. In fact, the real time information acquired by RFID sensors provides helpful data for farmers to adjust crop strategies at any time by taking into account the environmental condition alteration. Among smart agriculture paradigms, leaf sensing represents a new technology, which is used for the detection of plants’ health, such as water status [89,90]. An example of a low-cost and low-power system for leaf sensing using a plant backscatter sensor tag is presented in [90]. A schematic representation of the moisture monitoring RFID system and its practical implementation is reported in Figure 12. More in detail, a sensor measures the temperature differential between the leaf and the air, which is directly related to the plant water stress. A Morse code modulation was used for wireless communication with a low-cost receiver by backscattering RF signals from an 868 MHz carrier emitter. An example of an RFID soil moisture sensing system is reported in [91], which exploits two RFID tags on a pot with a low-pass filtered differential minimum response threshold (DMRT) to estimate the soil moisture level. The extensive measurement campaigns revealed high soil moisture detection accuracy, able to considerably improve the productivity. A passive UHF RFID tag for soil moisture and an environment temperature sensor for low cost agriculture applications was developed in [92]. The soil moisture sensor has been implemented by exploiting the capacitance variation of a passive inter digital capacitive, due to soil permittivity modification, while the environmental temperature was obtained by using an RFID chip with a temperature sensor. 

### 5.4. Automotive

RFID sensors for the automotive industry have exhibited a significant growth in the last few years, stimulated by the need for increasing the safety and reliability of vehicles, as well as to automate and improve its manufacturing and logistics processes. A recent example is given by the application of RFID technology for univocally identifying and localizing tires moving on conveyor lines [93]. The UHF RFID system is comprised of an RFID reading gate, located above a conveyor belt, that interacts with passive RFID tags hosted on the moving tires. RFID tag sensors can he zbe embedded in tires to provide information about the tires’ general condition, such as pressure and temperature. However, a big challenge for the adoption of RFID sensors in this case is represented by the ability to provide a robust and reliable implementation within the complex and harsh environment of a factory manufacturing tires. For example, the detuning effects of commercial RFID tags in tires is presented in [94]. Moreover, an example of flexible and stretchable RFID tag antennae for automotive tire applications are presented in [95]. The tag is manufactured with conductive textiles and embedded into polymer to improve its flexibility and bonding with the tire rubber. Other examples where RFID sensors can be useful for the automotive market are represented by a license plate RF identification system [96], intelligent parking [97], as well as child detection in vehicles [98,99,100,101,102] Some examples of RFID sensors applied in automotive applications are shown in Figure 13. 

### 5.5. Structural Health Monitoring

There are several applications which require security measures provided by structural health monitoring (SHM), such as the monitoring of bridges, railways and pipelines. In these applications, deterioration of and damage to these structures might occur during their operational lifespan [103]. For this reason, periodic visual inspections are performed, but they are proven to be unreliable, particularly when the structures are hard to access. Consequently, several non-destructive testing and evaluation (NDT & E) approaches, such as ultrasonic [104], pulsed eddy current (PEC) [105], and eddy current pulsed thermography (ECPT), were developed for monitoring defects in structures with good resolution, sensitivity, and reliability. Cabled large-scale sensor networks are very suitable for real time acquisition applications, because of their good performance. On the other hand, the use of these networks for manual data collection is rarely justified by the costs, installation difficulties, and maintenance [106]. Wireless sensor networks (WSNs) are cheaper and simpler to install by removing the electric wiring from traditional sensors. Spatial granularity is a crucial problem for possible upcoming applications. Battery-powered sensors are used for existing wireless sensing applications. Unfortunately, these sensors have reduced battery life and are more expensive than RFID sensors. Consequently, this restricts the granularity of their deployment and motivates low-cost, wireless, and passive sensor production for large-scale networks and applications for big data. Thanks to its low-cost, wireless, and “sensing-friendly” capabilities, radio frequency identification (RFID) technology can play a strategic role. The following paragraphs explain several examples of controlled parameters in structural health monitoring applications. 

*Strain detection*: to detect deformation or structural change that occurs in our surrounding infrastructure, strain sensors (gauges) are needed. In order to produce an effective strain sensor, researchers are looking for a material that, in response to a small applied strain, can display a significant structural change [107].

*Crack detection*: the development of fatigue cracks accounts for more than 50% of mechanical failures. The conventional crack sensing methods make use of lead wiring for data extraction, which is inefficient and costly for the positioning and maintenance of large lengths [108]. In [109], through the mutual-coupling between two patch antennas, it has been demonstrated that the backscattered phase can work as a sensing variable at a sub-mm resolution. A scheme of the crack detection system implemented with a couple of RFID tags is shown in Figure 14. In [110], a frequency signature-based chipless RFID is presented for metal crack detection and characterization operating in the ultra-wideband frequency. 

*Corrosion detection*: in the form of stress corrosion cracking (SCC) in susceptible metal components, the interaction of a corrosive environment and tensile stress (e.g., directly applied stresses or in the form of residual stresses) will cause failure. A thin film of oxides occurs in the early stages of corrosion and induces changes in the conductivity, permittivity and permeability of the metallic surface [111].

### 5.6. Space

Sensor RFIDs have also found applications on space platforms. Wire harnesses represent a considerable part of the overall mass; therefore, eliminating the wiring of the sensors is of the utmost importance for reducing the total mass of the vehicle, and thus the fabrication and launch costs. Moreover, a wireless system requires low-maintenance costs and offers an increase in reliability that is crucial, especially for long-term missions that can last years. An example of this is represented by the adoption of a RAIN RFID tag chip by the U.S. National Space Agency (NASA) [112]. The tag mounts a Monza X-8K Dura chip that can monitor temperature, CO2, and battery levels. A different approach for achieving the sensing function is provided by SAW RF tags [113]. In this case, two phase-matched SAW RF tags are coupled with a Van Atta antenna array. A combination of code diversity and time diversity has allowed for producing a set of 16 sensors that operate simultaneously in the field of view of the wireless sensor system. On the other hand, the Van Atta array provides a completely passive capability for tracking the probing direction and, at the same time, guarantees the beam-steering capability for properly retransmitting the encoded information. This configuration exhibits a remarkable read range, it can be employed for monitoring temperature and pressure, and it is robust enough to be deployed in a harsh environment. It is worth noting that, to reduce the weight of a satellite, the RFID reader itself is required to be compact in size and lightweight, as with the one proposed in [114] working within the 5725–5850 MHz frequency range.

### 5.7. Localization and Activity Monitoring

Even if, in these applications, RFIDs were not used as transducers for sensing purposes, RFID tags are still employed to extract useful information which exceeds the simple tracking of objects. RFID sensor networks are a promising approach for indoor activity monitoring [115,116,117]. A variety of experiments have been performed for tracking and tracing, indoors and outdoors. For example, in transportation systems, including buses, subways, and trains, the City of London implemented a system for outdoor data collection. The main goal was to use payments and access cards equipped with this connectivity technology to track users of the transport network. The level of usage, number of travelers, and public transport habits in the city, and knowledge of the stations, breakpoints, origin, and destination of network travel streams were calculated with the information collected. Indoor technologies have also been developed to consider citizens’ movement flows, such as museum visits. The collected data allowed for the identification of the visiting habits for the various areas and the identification of atypical behaviors [118].

Some of the mentioned drawbacks of a global positioning system (GPS) are addressed by RFID technology, as it does not require too much user cooperation, and reduces energy costs. However, its use for tracking does not provide precise positions, and the temporal position marks are restricted, like other wireless networks, by the locations of the signal reader antennas and the analysis of the readings’ received signal strength indicator (RSSI). The resolution of the trajectories followed by the users is less accurate than the resolution of a GPS. In addition, the outdoor aim of this technology is not comparable to GPS coverage. 

On the other hand, phase-based RFID localization methods allow for a better localization accuracy, since they are more robust to multipath propagation than the RSSI-based approaches. As an example, phase-based techniques have been used to localize moving tagged items on conveyor belts [119], robots [120,121], and drones [122]. Examples of robots and drones equipped with RFID readers for localization purposes are reported in Figure 15.

### 5.8. Wearable and Implantable RFID Sensors

One challenge is the design of wearable antennas that have high immunity to interactions with the human body, which may notably change the radiation diagram and degrade antenna efficiency. In active and semi-active architectures, as in the case of body-centric communication systems, the overall radiation performance is enhanced by additional battery-assisted electronics. In the case of passive tags instead, where the energy to produce the response comes from a remote unit, the antenna design is much more challenging. Embroidery techniques have been experimented with to fully integrate RFID tags inside clothes. E-textiles, conductive threads, and embroidery techniques proved to be suitable for the design of garment-integrated tags [123,124]. In order to enable on-body wireless connectivity, this new technology will include multi-functional everyday clothing integrated with wearable antennas, sensors, and power harvesting devices. The conductivity of the embroidered tag antennas mainly depends on the orientation of the sewn lines in relation to the current flow direction in the antenna system. Moreover, the conductivity of the embroidered structure is also determined by the electrical properties of the conductive thread. The thread used has a silver weight of 55 g/10,000 m, which assures good conductivity. Moreover, the conductivity and performance of the embroidered tag antennas is affected by the spacing between the sewn lines. An example of embroidered tags is shown in Figure 16a.

The maximum read range which is currently achievable with an RFID transmitter compliant with the regional power emission constraints is almost 5–6 m. Current link performances are already enough for tracking a person equipped with two tags over front/rear torso or over the arms within a regular size room [78].

In entertainment, healthcare, and medical applications, a wearable tag supplied with passive accelerometers can be applied on the arms for the tracking of human motion. In some common sleep disorders, such as restless leg syndrome and periodic limb movements, wearable tags with inertial switches have been shown to detect limb movements. Tags applied to the chest may also be useful to detect breath. Generally, wireless motion tracking systems may help to produce data to support diagnosis and track a patient’s behavior discreetly within a structure, and generate notifications about unusual activities, such as when the patient falls down or remains motionless for long periods [125].

RFID technology has been also demonstrated to be potentially useful for taking care of the human health-state internally by labeling body prostheses, sutures, stents, or orthopedic fixing. Each item could be monitored in real time or on demand by the IoT infrastructure for the ambitious goal of monitoring biophysical processes in evolution, such as tissue regrowth and prosthesis displacement. In this case, tags are inserted in the prosthesis, converting them into multi-functional devices capable of generating information in addition to providing the original medical functionality. In the design of implantable tags, the key challenge is to create a convenient communication link by using a reader power that complies with power emission regulation. Nowadays, the creation of an RFID connection with subcutaneous implants up to 0.1–0.5 m from the reader is now feasible, with promising applications for tracking some specific areas of the body, and vascular protheses. In order to automatically scan the health-state of a prosthesis without the direct involvement of the patient, a direct link to a reader-equipped door located at a distance of 0.5 m may be sufficient. On the other hand, deeper implants such as implants within the stomach remain a real challenge, even over the long term, for passive RFID systems under current power regulations [78]. 

A vascular stent implanted into an artery to restore natural blood flow after angioplasty is an example of near-subcutaneous protheses sufficient for RFID integration, as shown in Figure 16b. As presented in [126], the RFID tag features have been strongly incorporated into the stent geometry itself to act as a self-sensor by exploiting the dependency of the input impedance of the tag and back-radiation on the dielectric properties of the tissues in the immediate vicinity of the so-obtained STENTag. This makes it possible to detect the process of in-stent restenosis (ISR), e.g., diffuse proliferation of neointima early after implantation, or new atherosclerotic plaque, even at a distance of time, which might cause new occlusion of the vessel.

## 6. Commercial RFID Sensors

RFID technology has become extremely popular in recent years for both localization and sensing applications. This technology is widely used within the IoT paradigm, where the demand for low-power and low-cost wireless devices is increasing. From this perspective, RFID sensors respond to different needs of the IoT, as they are power efficient, small and easy to use. The use of innovative materials and manufacturing techniques, combined with increasingly advanced data collection techniques make RFID sensors even more appealing for IoT applications where sensing capabilities are required. Moreover, each RFID sensor has its own identification number, which makes it unique, making the collection of data from the sensor unambiguous. Finally, compared to other sensing and identification techniques, RFIDs are extremely advantageous, as they can be used in harsh environments, do not require a line-of-sight connection, and allow real-time collection of data from multiple sensors simultaneously. For these reasons, RFID sensors have raised great interest, not only in the world of scientific research, but also from industries. In fact, there are many companies that have heavily invested in RFID technology, such as TI, NXP, and ON Semicondutors.

The main manufacturers of ready-to-use RFID sensors are reported in Table 6, as well as some examples of RFID sensors that can be purchased on the market. Some commercial sensor tags configurations are reported in Table 7. Very often, these solutions can integrate with other sensors through an I2C interface, as in the case of TI RF430FRL152H. An interesting solution is proposed by NXP with the NTAG^®^ SmartSensor. These are single-chip solutions that combine NFC connectivity with autonomous sensing, data processing, and logging via the I2C interface. The data collected by NTAG SmartSensor are uploaded into the cloud via NFC using a smartphone and a dedicated app. The NTAG SmartSensors are temperature-calibrated, which is an important feature for tracking temperature-sensitive products such as vaccines or wines. By using the I2C interface, the NTAG SmartSensor can also monitor conditions such as humidity, tilting, shocks, and vibrations. Another application of NXP NTAG is smart agriculture (Figure 17). The NTAG^®^ SmartSensor Figure 17a is placed in the plant, as shown in Figure 17b. Another example of a commercial RFID sensor is HYGRO-FENIX-RM of Farsens (Donostia-San Sebastian, Spain), which is an electronic product code (EPC) class 1 generation 2 (C1G2) RFID tag based on Farsens’ battery-less sensor technology. The tag contains an ambient temperature sensor and a relative humidity sensor, which are compatible with commercial UHF RFID readers (EPC C1G2). The battery-less resistance meter can communicate over 5 m with a 2W ERP setup. Farsens also produces other passive sensor tags, such as the EVAL01-Kineo-RM, which is a UHF RFID battery-free orientation sensor tag. It features an LIS3DH 3-axis accelerometer from ST Microelectronics with a range between ±4 g and an accuracy of ±40 mg. Other passive sensors are proposed by Farsens to sense parameters such as RF field (EVAL02-Photon-R) and ambient light intensity (EVAL01-Spectre-RM). 

It is also worth mentioning the use of RFID sensors for the moisture intrusion detection system for vehicle assembly lines. Water leakage has been one of the main problems in the auto industry for years. Indeed, water leaks might cause mold growth and damage to the expensive electronic components of motor vehicles, thus reducing the quality of the fabrication process. Traditional manual inspection methods rely on visual inspection. Consequently, this approach does not detect leaks located in inaccessible areas, or small leaks. ON Semiconductor, Inc. (Phoenix, AZ, U.S.A.) and RFMicron, Inc. (Austin, TX, USA) have solved this problem with a two-fold leak detection solution [127]. The first part utilizes thin and low-profile moisture sensors that are placed in the vehicle without affecting or displacing other components or trim pieces (Figure 18a). The second part is a system to read these sensors and then aggregate that sensor data to determine where leaks are located (Figure 18b). Battery-free wireless moisture sensors and a highly capable processing device mounted directly onto moving assembly lines are included in the RFM5126 water leak detection system. A wireless communication between the antennas mounted on a system portal and the sensors located on the vehicles is established. When the vehicles drive through the portal, the system collects the data from the sensors. In order to decide whether and where any leaks might be present, the device processes the sensors’ data. The identification of the water leaks allows for a reduction in repair costs and saves time, so that rework teams do not spend their time searching for leaks.

## 7. RFID Readers

RFID systems can operate at different frequency bands: low-frequency (LF, 125–134 KHz), high-frequency (HF, 13.56 MHz), ultra-high-frequency (UHF, 866–928 MHz), and microwave (MW, 2.45 GHz and 5.8 GHz), as summarized in Table 8. The specific operating frequency is chosen on the basis of the application requirements. For example, LF and HF are typically used for short-range applications (e.g., of a few centimeters), since the wireless power transmission is based on the inductive coupling between two coils. The magnetic field received by the tag coil induces a current which activates the chip. The chip impedance is then varied based on the EPC stored inside the chip. Such a load variation induces a modulation of the backscattered signal. The latter is then collected at the reader side and decoded. On the other hand, UHF RFID systems are typically used for long-range applications, since they are based on electromagnetic coupling and wave propagation. The backscattered electromagnetic signal is then received by the reader antenna and converted to an electrical signal. If the received power is higher than the reader sensitivity, then the interrogated tag can be properly detected, and the unique ID (EPC) can be stored.

A RFID reader is powered by a battery or from an external power source, and generates an interrogation signal centered at the specific operating frequency. The signal is then emitted by the RFID reader antenna after an electrical-to-electromagnetic conversion. The carrier is then received by tag antennas placed at a certain distance from the reader antenna. If the power received by the transponder is higher than the chip sensitivity, then the chip is powered up and sends the unique ID stored in it. Specifically, the interrogation signal is backscattered by the tag after a load modulation, which causes a little shift in the operating frequency, as schematically represented in Figure 19.

RFID readers can be classified into three main typologies:-**Fixed or desktop reader**. In applications where the available space is not limited, fixed readers can be used, which guarantee the best performance. The typical size is smaller than 30 cm × 30 cm and the cost is approximately in the range of 800–1200 USD. Some commercial readers are listed in Table 9, also including the operating frequency and main performance parameters. In general, readers can be connected to up to four different external antennas, and a time division approach is used to select the specific radiating element.-**Integrated readers**. Several commercial solutions are integrated systems, where both the reader and the antenna are embedded in the same device (Table 10). This represents a compact solution for most of the applications in the retail or pharmaceutical industries. However, differently from fixed readers, the antenna cannot be changed, thus the RFID system cannot be fully designed on the basis of the operative scenario. In HF RFID commercial integrated readers, the radiating element is represented by a coil, properly tuned at 13.56 MHz. On the other hand, in UHF RFID readers, different typologies of antennas are integrated to guarantee reliable detection when the tag is placed both on the reader case and at a longer distance [128]. Integrated reader’s typical cost is in the range of 700–1200 USD.-**Portable/handheld or wearable readers**. Handheld readers represent a lightweight and easy-to-carry solution, which can be used by a human operator for scanning tagged items placed in different positions (Table 11). In this framework, portable reader antennas must be integrated in a compact space, thus they are required to be lightweight, low profile, and compact in size. Consequently, the read range is limited to up to a few meters. Furthermore, the proximity of the human hand represents an important aspect to take into account during the antenna design process, because the dielectric properties of human tissues may significantly affect the antenna’s performance. Handheld readers’ typical cost is in the range of 200–900 USD.

It is worth mentioning that software defined radio (SDR) readers have been also designed and described in the scientific literature [129,130], but they are not commercially available yet.

Most commercial readers are not able to directly provide information on the data collected from the sensor tag. Typically, only a few parameters are provided on the graphical user interface, such as:Tag EPC,RSSI (received signal strength indicator) of each detected tag, giving an estimation of the quality of the detection signal,Antenna port. Since fixed readers can support up to four external antennas, information on which antenna detected a tag is fundamental,Time stamps of the first and last reads of each tag ID.

All of this information can be automatically sent to a management system/software, or collected into a logfile (e.g., in .txt format) for post processing operations. Not all commercial RFID readers provide information on the phase of the received backscattered signal. This parameter is proportional to the distance between the reader antenna and the tag, and it can be used, for example, in phase-based localization systems and angle-of-arrival (AoA) estimations if multiple antennas are used at the reader side. Among other commercial readers that provide phase information, the Impinj Speedway Revolution R420 UHF-RFID reader can be mentioned.

Sometimes, software modifications and add-ons are needed at the reader side to read specific registers of the RFID chip, thus providing values from the sensors (e.g., temperature or humidity). For example, Farsens demo software patch is available online for testing battery-free sensor tags. However, a PC version of such software is only compatible with few UHF RFID readers, such as the Impinj Speedway (RX20), ThingMagic M6, Alien ALR-9900, Zebra FX9500, or Nordic ID Sampo S1/2.

On the other hand, chipless RFID readers are not yet available on the market. However, they have been proposed in the scientific literature, and some configurations are listed in Table 12. In general, a chipless RFID reader is a type of radar that can be designed with either a frequency domain or a time domain approach [131]. The frequency domain strategy expects the transmission of a harmonic, which varies in frequency within the operative bandwidth of the tag. The reader architectures used in the literature are based on either the stepped-frequency continuous wave (SFCW) or FMCW concepts. Both configurations are equipped with a voltage-controlled oscillator (VCO) in transmission, whose center frequency is settled by a control unit [131]. A directional coupler at the VCO output is used as an input for the demodulator. A reader founded on the time domain methodology transmits a sub-nanosecond pulse toward the tag, and measures the backscattered signal from the tag. This technique is used in impulse radio UWB (IR-UWB) technology, where the reader is composed of a pulse generator and a time domain receiver. The transmitted pulse should be compliant with the UWB regulation masks. The power spectral density (PSD) over megahertz is calculated over a period of time much higher than 1 ns (1 ms for the FCC, 1 ms–1 μs for the ETSI) [131]. 

## 8. Conclusions

A comprehensive overview of RFID sensing technology has been presented. The advantages of RFID sensors with respect to classical battery-equipped sensor nodes are highlighted. Then, a detailed classification of RFID sensors is provided, and the different architectures are described and discussed. The main areas of application of RFID sensors have been presented. Finally, some commercial RFID sensing solutions are summarized, as well as the readers available on the market. 

## Figures and Tables

**Figure 1 sensors-21-03138-f001:**
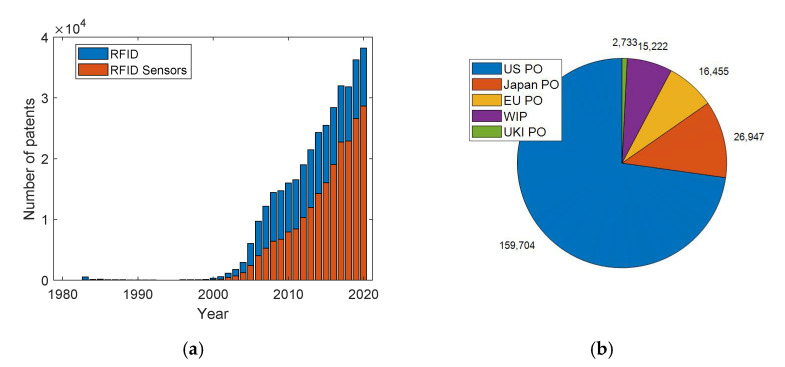
Number of (**a**) patents indexed on the Scopus database in February 2021 according to the keyword ‘RFID’ and the keyword ‘RFID Sensor’. (**b**) Number of patents issued by the main patent offices. Acronyms: United States Patent and Trademark Office (US PO), Japan Patent Office (Japan PO), European Patent Office (EU PO), World Intellectual Property Organization (WIP), United Kingdom Intellectual Property Office (UKI PO). Source: Scopus.

**Figure 2 sensors-21-03138-f002:**
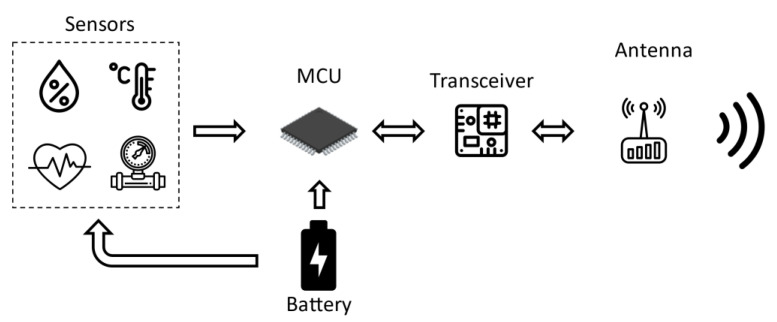
Simplified sketch of classical wireless sensor architecture.

**Figure 3 sensors-21-03138-f003:**
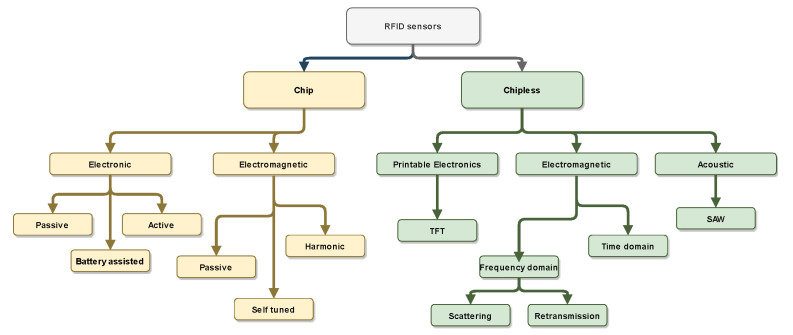
Classification of RFID sensors. Acronyms: SAW (surface acoustic wave), TFT (thin film transistor).

**Figure 4 sensors-21-03138-f004:**
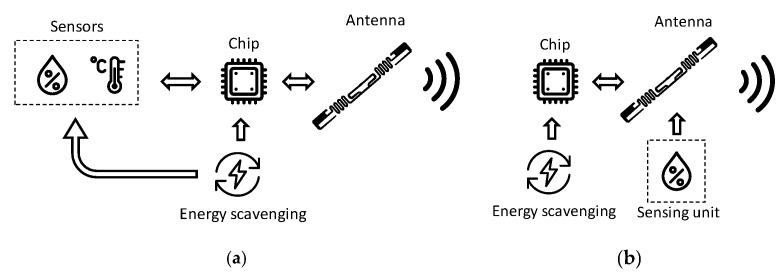
RFID-based sensor topology: (**a**) electronic RFID sensor and (**b**) electromagnetic RFID sensor architecture.

**Figure 5 sensors-21-03138-f005:**
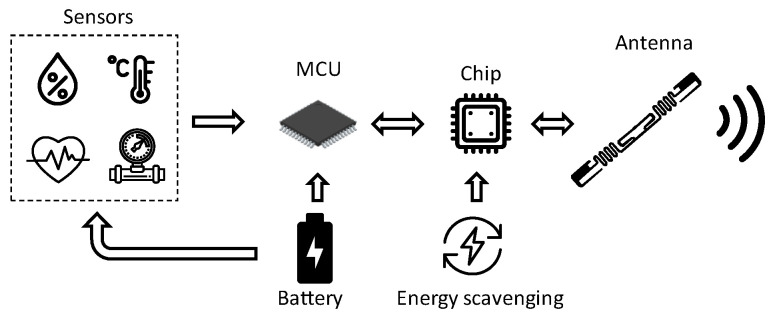
RFID tags with battery-assisted sensors.

**Figure 6 sensors-21-03138-f006:**
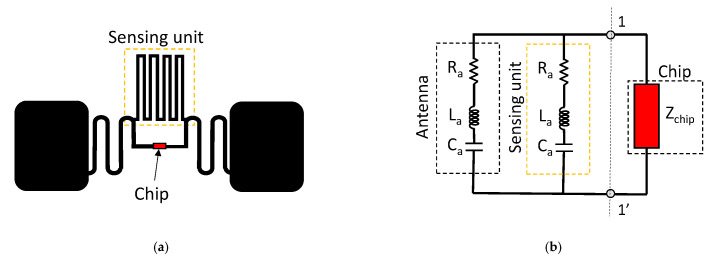
(**a**) Equivalent circuit model of an electromagnetic RFID sensor. (**b**) Possible layout of the sensor. Working principle of frequency domain RFID sensors based on (**c**) capacitive or (**d**) resistive transduction mechanisms.

**Figure 7 sensors-21-03138-f007:**
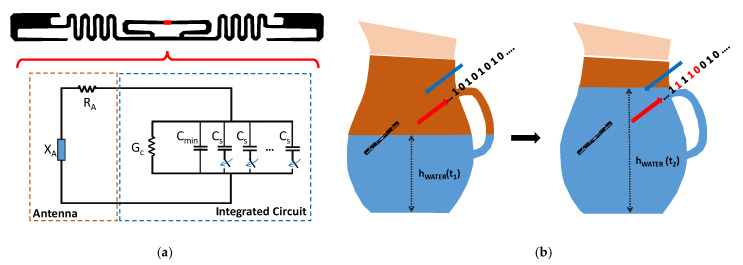
(**a**) Equivalent circuit of an RFID tag equipped with a self-tuning chip and (**b**) an example of the exploitation of the autotuning feature for indirect sensing.

**Figure 8 sensors-21-03138-f008:**
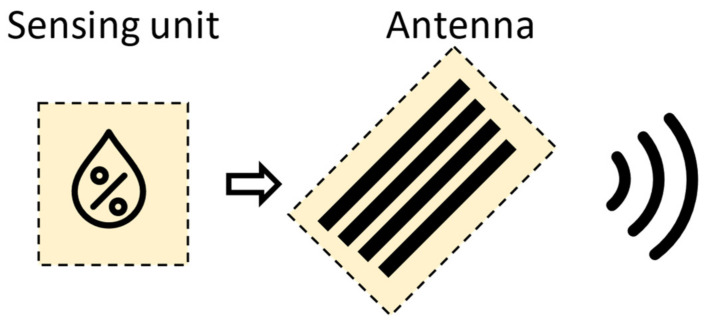
Chipless RFID sensor architecture and possible layout of a chipless RFID sensor.

**Figure 9 sensors-21-03138-f009:**
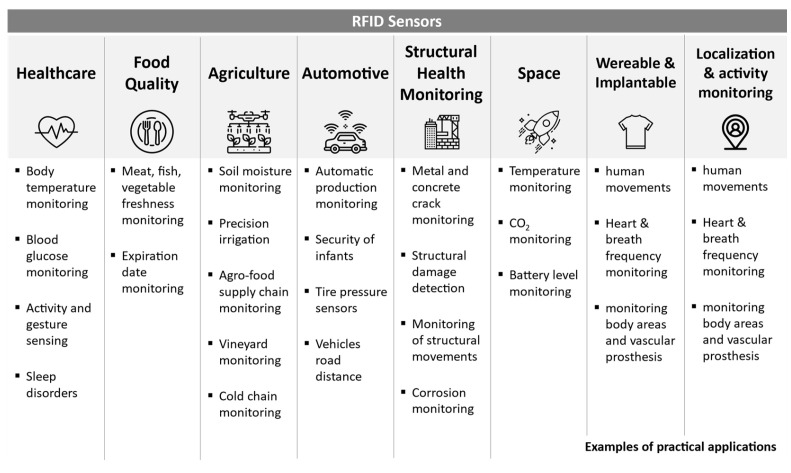
RFID sensors: application fields and examples of practical applications.

**Figure 10 sensors-21-03138-f010:**
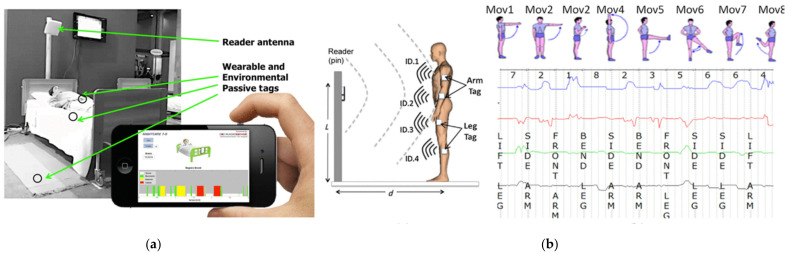
(**a**) Ambient intelligence system aimed at taking care of night sleep involving RFID tags placed over the body and in the surrounding environment. (**b**) Set-up for the classification of arm and leg gestures by passive RFID and examples of RFID backscattered patterns. Reproduced from [78], © 2014 IEEE.

**Figure 11 sensors-21-03138-f011:**
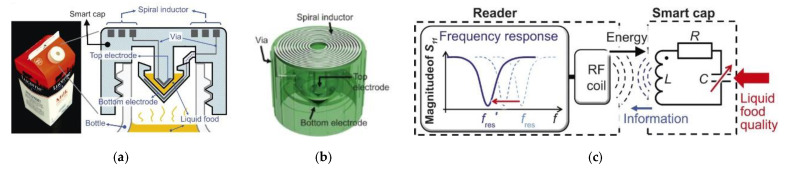
(**a**,**b**) A wireless passive sensor demonstration of a “smart cap”, containing the 3D-printed LC-resonant circuit. The degradation of the liquid food inside the liquid package can cause changes in the dielectric constant and a shift in the resonance frequency of the LC circuity. (**c**) A wireless inductive reader is used to monitor the signals in real time. Reproduced from [86].

**Figure 12 sensors-21-03138-f012:**
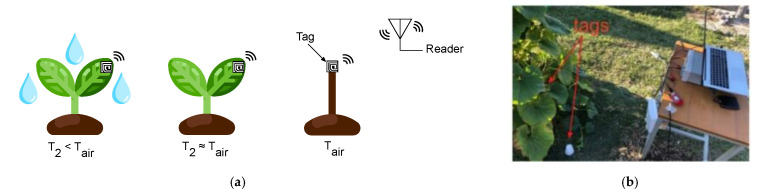
Scheme of the RFID-based autonomous leaf-compatible temperature sensing system proposed in [89], © 2019 IEEE, and experimental setup. (**a**) working principle of the RFID system. (**b**) A picture of the realized system.

**Figure 13 sensors-21-03138-f013:**
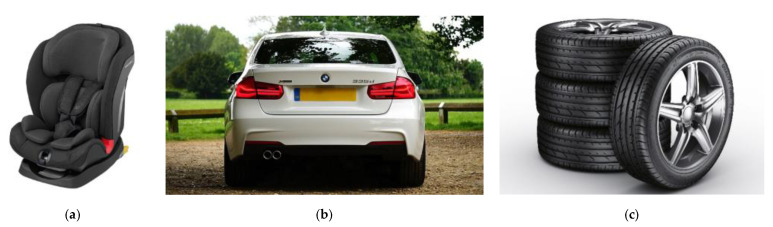
Possible employment of as RFID system for preventing child abandonment inside cars (**a**), RFID on a license plate (**b**), RFID inside tires (**c**).

**Figure 14 sensors-21-03138-f014:**
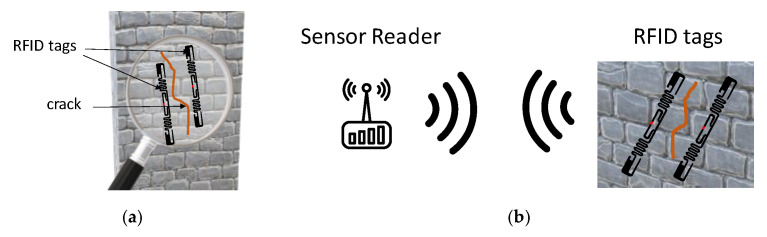
Principle of (**a**) an RFID sensor tag couplet for the detection of cracks on concrete structures [109], © 2015 IEEE; (**b**) an RFID chipless RFID sensor system for crack detection and characterization on a metallic structure.

**Figure 15 sensors-21-03138-f015:**
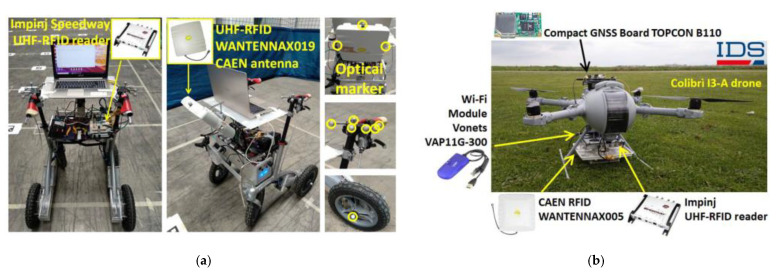
(**a**) Robotic wheeled walker equipped with the Impinj Speedway Revolution R420 UHF-RFID reader, the reader antenna, and optical markers used for the measurement campaign in [120], © 2021 IEEE; (**b**) Colibrì I3-A drone by IDS, equipped with a GNSS module and the UHF-RFID reader and antenna used in [122], © 2019 IEEE.

**Figure 16 sensors-21-03138-f016:**
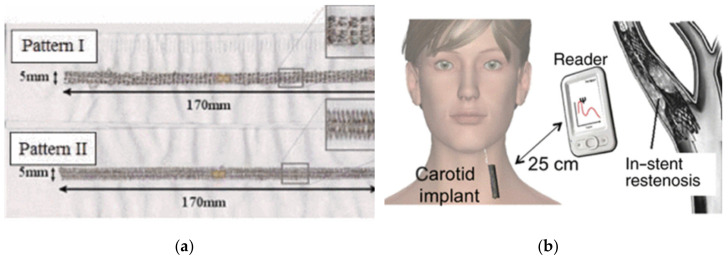
(**a**) Embroidered dipoles with pattern I and pattern II. The application of UHF RFID technology for monitoring. Reproduced from [124], © 2013 IEEE. (**b**) The in-stent restenosis inside a carotid stent and a prototype of a STENTag. Reproduced from [78], © 2014 IEEE.

**Figure 17 sensors-21-03138-f017:**
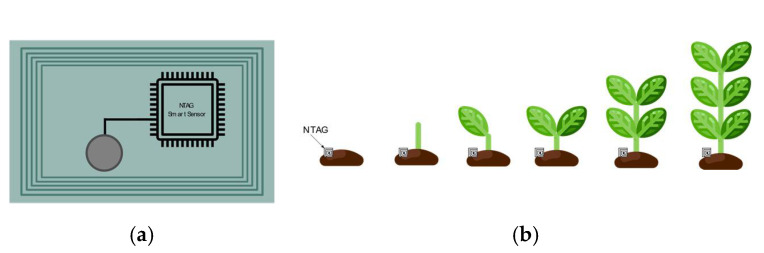
NTAG^®^ SmartSensor for smart agriculture: (**a**) pictorial representation of the NTAG^®^; (**b**) application of the sensor in the plant.

**Figure 18 sensors-21-03138-f018:**
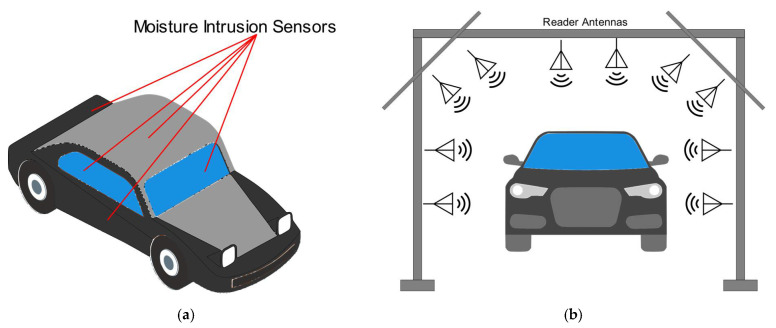
RFID sensors for the moisture intrusion detection system: (**a**) Moisture sensors placed directly on the vehicles’ metal chassis; (**b**) Interrogation setup.

**Figure 19 sensors-21-03138-f019:**
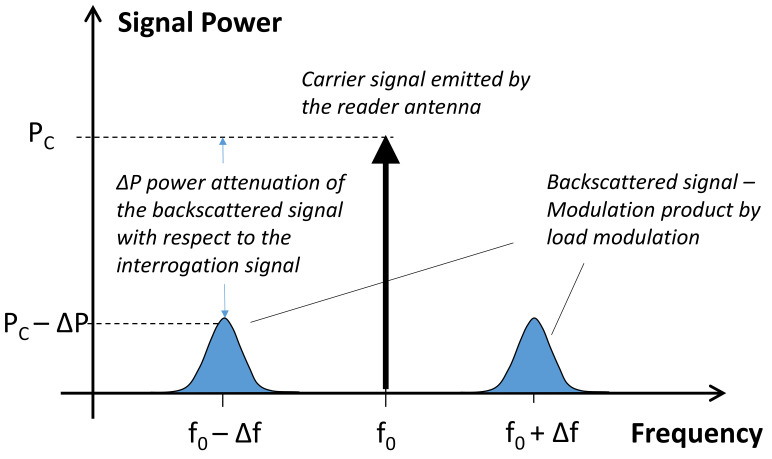
Load modulation creates two sidebands around the transmission frequency f_0_ of the reader. The actual information is carried in the sidebands of the two subcarrier sidebands, which are themselves created by the modulation of the subcarrier.

**Table 1 sensors-21-03138-t001:** Industrial IoT Technologies.

	Wired Sensors	Wireless Sensors	RFID Sensors	Chipless RFID Sensors
Installation cost	High	Low	Low	Low
Maintenance cost	Low	High	Low	Low
Hardware cost	High	Medium	Low	Low
Scalability	Low	High	High	High
Sensor accuracy	High	High	Medium	Low

**Table 2 sensors-21-03138-t002:** Comparison between main chipless RFID technologies.

	Printed Electronics	EM Scattering	SAW
Read range	<10 cm	<1 m	<10 m
Working band	13 MHz	10–100 MHz/3–8 GHz	2.45 GHz
Maturity level	Medium	Medium	High
Temperature range		−100°–300°	−100°–300
Standard	HF-RFID	No	Some
Cost	NA	Very low (<1 cent)	Low (<10 cents)
Anticollision	Good	Low	Good
Read speed	Low	Low	High
Threshold power	Yes	No	No
Bits	High	Low	High
Rewritable	Yes	No	No
Market available	No	No	Yes
Integration with packaging	Yes	Yes	No
Sensing	Yes (various)	Yes (various)	Yes (temperature)

**Table 4 sensors-21-03138-t004:** Sensing polymers and nanomaterials.

Category	Material Name	Sensing
Polymers	Stanyl^®^TE200F6	Temperature
	Polyamide kapton	Humidity
	Polyvinyl alcohol (PVA)	Humidity
	Poly (2-Hydroxyethyl methacrylate)	Humidity
	Polydimethylsiloxane (PDMS)	pH, pressure
	Polymethyl methacrylate (PMMA)	Humidity, methanol
	PEDOT:PSS (poly(3,4-ethylenedioxythiophene) polystyrene sulfonate)	Temperature, humidity, pH
	cellulose	Chemical sensors (humidity, ion, gas)
Nanomaterials	Metal and metal oxide semiconductor (MOS): SnO2, ZnO and TiO2	Gasses
	Nanoparticle-based inks	Strain
	Carbon nanotubes (CNTs)	Gas, temperature, pressure/mechanical strain
	Graphene	Strain, temperature, biological, gasses (NO2, NH3 and CO)
	Barium strontium titanate (BST)	Temperature
	Zeolite Material	Ammonia
	Paper	Humidity

**Table 5 sensors-21-03138-t005:** Performance comparison and utilization of printed electronics and silicon ICs in flexible hybrid electronics (FHE) systems [73].

Performance Parameters	Printed Electronics	Silicon ICs
Charge carrier mobility (cm^2^ V^−1^ s^−1^)	~1 (organics)	~1000
Switching speed (MHz)	~1	~5000
Operating voltage (V)	~10	~1
Lifetime (yr)	~0.1	~10
Substrate softness—elastic modulus-1 (Gpa^−1^)	~2.5–1 (plastics)	~180–1

**Table 6 sensors-21-03138-t006:** Principal RFID sensors manufacturers. The websites have been accessed on 1 March 2021.

Company Name	Website	Sensor
AMS	https://ams.com/wireless-sensor-tags-interfaces	Temperature/integration with external sensors
Axzon (RFmicron)	https://axzon.com/sensors/	Temperature, moisture
Farsens	http://www.farsens.com/en/products/battery-free-rfid-sensors/	Temperature, humidity, force/strain, Pressure, LED/RF field detection, Switch/relay monitoring, orientation, light, magnetic field
Infratab	https://www.infratab.com/rf-sensor-solutions	food freshness
Melexis	https://www.melexis.com/en/products/communicate/nfc-rfid-products	Temperature/integration with external sensors
NXP	https://www.nxp.com/products/rfid-nfc/nfc-hf/ntag/ntag-smartsensor:NTAG-SMART-SENSOR	Temperature
ON Semiconductor	https://www.onsemi.com/products/sensors/battery-free-wireless-sensor-tags	Temperature, moisture
Phase IV	https://www.phaseivengr.com/products/rfid-sensors/	Temperature, moisture, pressure, strain
PowerCast	https://www.powercastco.com/products/rfid-sensor-tags/	Temperature, humidity, light
RadioForce	https://www.radioforce.net/en/products/sensors	Temperature, humidity, water detection, Motion, vibrations, pressure
Silent Sensors	https://www.silentsensors.com/	Tire conditions
Smartrac	https://www.smartrac-group.com/sensor-inlays-and-tags.html	Temperature, moisture
Texas Instruments	https://www.ti.com/wireless-connectivity/other-wireless-technologies/overview/nfc-rfid.html	Temperature + SPI/I2C external sensor
PST Sensors	https://www.pstsensors.com/wireless.htm	Temperature

**Table 7 sensors-21-03138-t007:** Examples of commercial RFID sensors.

Tags							
Manufacturer	Sensor Model	Sensing Function	RFID Band	Accuracy	Type	Size [mm^3^]	Unit Price
ON Semiconductor	SPS1M001FOM	Moisture	UHF	N/A	passive	165.7 × 20 × 5	$7.3
ON Semiconductor	SPS1T001PET	Temperature	UHF	±0.5° @ 30°	passive	101.60 × 31.75 × 5	$3.99
Axzon (RFmicron)	RFM2110	Moisture	UHF	N/A	passive	79.5 × 19.2 × 2.1	$5.7
Farsens	EVAL01-Hygro-Fenix-RM-DKWB	Temperature and relative humidity	UHF	±1 °C [0 °C–60 °C] ±3.5% rH [20% rH–80% rH]	passive	137 × 16 × 10	$58
Farsens	EVAL02-Kineo-RM-DKSWB	Orientation (3-axis accelerometer)	UHF	range ±4 g acceleration accuracy ±40 mg	passive	137 × 17 × 10	$42
Farsens	EVAL02-Photon-R	RF field detection	UHF	ON/OFF	passive	160 × 27.5 × 10	$42
Farsens	EVAL01-Spectre-RM	Ambient light	UHF	Optical spectrum: 300 nm to 1000 nm Range: 1.2 nW/cm^2^ to 10 mW/cm^2^	passive/BAP	137 × 16 ×10	$60
Farsens	EVAL01-Zygos-RM	Force/strain	UHF	Compression load range: 0 to 5 kg, Compression load accuracy: ±0.05 kg	passive/BAP	95 × 95 × 45 137 × 16 × 10	$48
**Chip**							
**Manufacturer**	**Sensor Model**	**Sensing Function**	**RFID Band**	**Accuracy**	**Type**	**size [mm^3^]**	**Unit Price**
AMS	AS39513	Temperature + external resistive sensor	HF	Default range: −20 °C to 55 °C ±0.5 °C over −20 °C to 10 °C	passive/BAP	Chip: 2.5 × 2.5 × Δ	$1.54
Melexis	MLX90129	Temperature	HF	±2.5 °C with calibration	passive/BAP	Chip 6.4 × 6.4 × 1.1	~$3
NXP	NTAG^®^ SmartSensor	Temperature/moisture	HF/UHF	±0.5 °C from −40 °C–0 °C ±0.5 °C from 40 °C–85 °C	BAP	Chip: 4 × 4 × 0.85	$1.5/3
Texas Instruments	RF430FRL152H	Temperature + SPI/I2C external sensor	HF	N/A	passive/BAP	Chip: 4 × 4 × 0.75	$3.8

**Table 8 sensors-21-03138-t008:** RFID operating frequencies and standards.

	LF	HF	UHF	Microwave
Freq. Range	125–134 KHz	13.56 MHz	865–868 MHz (ETSI) 902–928 MHz (FCC)	2.45–5.8 GHz
Read Range	10 cm	1 m	0–10 m	1 m
Market share	74%	17%	6%	3%
Coupling	Magnetic	Magnetic	Electromagnetic	Electromagnetic
Existing standards	11,784/85, 14,223	18,000-3.1, 15,693, 14,443 A, B, and C	EPC C0, C1, C1G2, 18,000-6	18,000-4
Applications	Smart card, ticketing, animal tagging, access, laundry	Small item management, supply chain, anti-theft, library, transportation	Transportation vehicle ID, access/security, large item management, supply chain	Transportation vehicle ID, access/security, item management, supply chain

**Table 9 sensors-21-03138-t009:** Examples of commercial fixed RFID readers. The websites have been accessed on 1 March 2021.

MODEL(Manufacturer)	Picture	Operating Frequency Band	Number of Antenna Ports	Website Link
Quattro-R4321P(CAEN RFID s.r.l.)	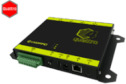	865–868 MHz (ETSI UHF)902–928 MHz (FCC UHF)	4	https://www.caenrfid.com/en/products/quattro-r4321p/
Impinj R700 reader (Impinj)	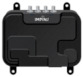	865–868 MHz (ETSI UHF)902–928 MHz (FCC UHF)	4	https://www.impinj.com/products/readers/impinj-r700
Impinj Speedway (Impinj)	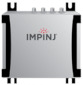	865–868 MHz (ETSI UHF)902–928 MHz (FCC UHF)	4	https://www.impinj.com/products/readers/impinj-speedway
FX7500 (ZEBRA)	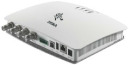	865–868 MHz (ETSI UHF)902–928 MHz (FCC UHF)	4	https://www.zebra.com/it/it/products/rfid/rfid-readers/fx7500.html
FX9600 (ZEBRA)	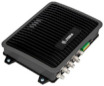	865–868 MHz (ETSI UHF)902–928 MHz (FCC UHF)	4	https://www.zebra.com/it/it/products/rfid/rfid-readers/fx9600.html

**Table 10 sensors-21-03138-t010:** Examples of commercial integrated RFID readers. The websites have been accessed on 1 March 2021.

MODEL(Manufacturer)	Picture	Operating Frequency Band	Nominal Read Range	Website Link
IceKey HF (Tertium Technology)	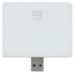	13.56 MHz (HF)	10 cm	https://www.tertiumtechnology.com/products/desktop-hf/
Atlantis Desktop HF/NFC (Tertium Technology)	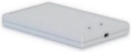	13.56 MHz (HF)	10 cm	http://www.sensorid.it/products/atlantis-desktop.html
IceKey UHF (Tertium Technology)	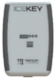	865–868 MHz (ETSI UHF)902–928 MHz (FCC UHF)	1.2 m	https://www.tertiumtechnology.com/products/desktop-uhf/
Slate-R1260E/UUSB Desktop RAIN RFID (CAEN RFID)	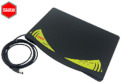	865–868 MHz (ETSI UHF)902–928 MHz (FCC UHF)	Short- to medium-range	https://www.caenrfid.com/en/products/r1260-slate/
Discovery Desktop (SENSOR ID)	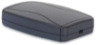	865–868 MHz (ETSI UHF)902–928 MHz (FCC UHF)	50 cm	http://www.sensorid.it/products/discovery-desktop.html
Discovery Gate PI (SENSOR ID)	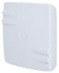	865–868 MHz (ETSI UHF)902–928 MHz (FCC UHF)	8 m	http://www.sensorid.it/products/discovery-gate-pi.html

**Table 11 sensors-21-03138-t011:** Examples of commercial handheld RFID readers. The websites have been accessed on 1 March 2021.

MODEL (Manufacturer)	Picture	Operating Frequency Band	Nominal Read Range	Website Link
BLUEBERRY HS HF (Tertium Technology)	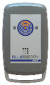	13.56 MHz (HF)	6 cm	https://www.tertiumtechnology.com/products/desktop-hf/
C71 UHF RFID Reader (CHAINWAY)	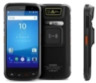	13.56 MHz (NFC)865–868 MHz (ETSI UHF)902–928 MHz (FCC UHF)	2–4 cm (NFC)3 m (UHF RFID)	https://www.chainway.net/Products/Info/41
BLUEBERRY HS UHF (Tertium Technology)	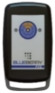	865–868 MHz (ETSI UHF)902–928 MHz (FCC UHF)	30 cm	https://www.tertiumtechnology.com/prodotto/blueberry-hs-uhf/
Discovery Mobile UHF (SENSOR ID)	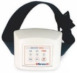	865–868 MHz (ETSI UHF)902÷928 MHz (FCC UHF)	50 cm	http://www.sensorid.it/products/discovery-mobile-uhf.html
qID-R1240IE/IUWearable Bluetooth RAIN RFID/BARCODE (CAEN RFID)	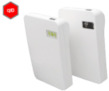	865–868 MHz (ETSI UHF)902–928 MHz (FCC UHF)	1.5 m	https://www.caenrfid.com/en/products/r1240ieiu-qid/
qIDmini-R1170IKeyfob Bluetooth RAIN RFID (CAEN RFID)	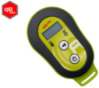	865–868 MHz (ETSI UHF)902–928 MHz (FCC UHF)	90 cm	https://www.caenrfid.com/en/products/r1170i-qidmini/
Culla RFID UHF RFD2000 (ZEBRA)	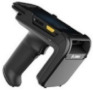	865–868 MHz (ETSI UHF)902–928 MHz (FCC UHF)	6 m	https://www.zebra.com/it/it/products/rfid/rfid-handhelds.html

**Table 12 sensors-21-03138-t012:** Performance and cost comparison of chipless RFID tag readers [132].

Reference	Operating Bandwidth	Reader System	Cost (Approx.)	Background Calibration
[43]	5–20.5 GHz	VNA-based	>$3000	yes
[132]	2–2.5 GHz	VCO	$120	no
[133]	4–8 GHz	IR-UWB	<$650	no
[134]	3.1–10.6 GHz	IR-UWB	<$2500	no
[135]	2.4–3.4 GHz	VCO	<$1000	yes

## Data Availability

Not applicable.

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
