# Peer review of "A Review of RFID Sensors, the New Frontier of Internet of Things"

_sensors, 2021, doi:10.3390/s21093138_

Round 1

Reviewer 1 Report

Authors showed the RFID sensor review paper for wireless sensing technology. Authors also showed chipless RFID sensing technology. Authors showed good classifications of RFID sensors. Tables and Figures for RFID sensor technology are well organized to describe the technologies. However, authors missed some references and need to improve the Figures quality. Therefore, the manuscript can be minor revision if authors answer the following comments.

1. Please correct wi-fi to Wi-Fi.
2. Authors need to add the reference (In case of battery assisted RFID sensors or battery-assisted passive mode, a simple circuit is built around the memory chip, thus enabling the chip to switch to a local energy-assisted mode only when it senses a certain stimulus.)
3. Authors need to add the reference (In particular, the number of patents published every~).
4. Authors need to add the reference (The patents classified for the registration office are reported in Figure 1 (b) to highlight the ~).
5. Authors need to add the reference (Finally, systems based on battery-powered communication still cost more than US$5 a unit, with no clear commercial or tech nical solution to achieve a cost lower than a dollar.)
6. Authors need to add the reference (Electronic sensor tags have the peculiarity of separating the sensing functions from the~).
7. Authors need to add the reference (Active RFID systems generally work at 433 MHz or 2.45 GHz, but 433 MHz~).
8. Authors need to add the reference (RFID tags can cost from $20 to $100+ depending on the tag specifications~)
9. Authors need to add the reference (When the reader interrogates the BAP tag, the embedded battery is turned on as well as~)
10. Authors need to add the reference (The Monza X-2K RFID chip provides a performance boost via a ‘battery-assisted~)
11. Authors need to add the reference (A possible layout of the RFID sensor is shown in Figure 6(a), where a dipole antenna~).
12. Authors need to check MDPI reference styles.
13. Figure 7 b labels are small.
14. Table 3 fonts need to be corrected.
15. Figure 10 labels are small.
16. No author contribution information.
17. Authors need to provide data availibity. Please check MDPI styles.
18. Figure 15 labels are not clearly to be shown.
19. Figure 14 labels are not clear.
20. Figure 12 label is small.

Author Response

Authors showed the RFID sensor review paper for wireless sensing technology. Authors also showed chipless RFID sensing technology. Authors showed good classifications of RFID sensors. Tables and Figures for RFID sensor technology are well organized to describe the technologies.

Thanks for the positive assessment of the paper.

However, authors missed some references and need to improve the Figures quality. Therefore, the manuscript can be minor revision if authors answer the following comments.

  1. Please correct wi-fi to Wi-Fi.

Thanks for the suggestion. We have corrected.

  1. Authors need to add the reference (In case of battery assisted RFID sensors or battery-assisted passive mode, a simple circuit is built around the memory chip, thus enabling the chip to switch to a local energy-assisted mode only when it senses a certain stimulus.)

Reference has been added.

  1. Authors need to add the reference (In particular, the number of patents published every~).
  2. Authors need to add the reference (The patents classified for the registration office are reported in Figure 1 (b) to highlight the ~).

The statements are based on the database Scopus. We think it is not necessary to add a reference to Scopus. We have added the source of the data in the figure caption.

  1. Authors need to add the reference (Finally, systems based on battery-powered communication still cost more than US$5 a unit, with no clear commercial or tech nical solution to achieve a cost lower than a dollar.)

A reference has been added.

  1. Authors need to add the reference (Electronic sensor tags have the peculiarity of separating the sensing functions from the~).

A reference has been added.

  1. Authors need to add the reference (Active RFID systems generally work at 433 MHz or 2.45 GHz, but 433 MHz~).
  2. Authors need to add the reference (RFID tags can cost from $20 to $100+ depending on the tag specifications~)

A reference has been added.

  1. Authors need to add the reference (When the reader interrogates the BAP tag, the embedded battery is turned on as well as~)

A reference has been added.

  1. Authors need to add the reference (The Monza X-2K RFID chip provides a performance boost via a ‘battery-assisted~)

A reference has been added.

  1. Authors need to add the reference (A possible layout of the RFID sensor is shown in Figure 6(a), where a dipole antenna~)

A reference has been added.

  1. Authors need to check MDPI reference styles.

Reference style for mdpi has been imported in Zotero.

  1. Figure 7 b labels are small.

Font size of both figure 7a and figure 7b has been increased.

  1. Table 3 fonts need to be corrected.

The font of table 3 has been corrected.

  1. Figure 10 labels are small.

The figure, which is taken from another paper has been enlarged to make the labels more readable.

  1. No author contribution information.

Author contributions have been updated.

  1. Authors need to provide data availability. Please check MDPI styles.

There is not data associated with this paper as it is a review paper.

  1. Figure 15 labels are not clearly to be shown.

The figure, which is taken from another paper has been enlarged to make the labels more readable.

  1. Figure 14 labels are not clear.

Figure 14 has been redrawn.

  1. Figure 12 label is small.

Font size of labels of Fig. 12 has been increased.

Reviewer 2 Report

Hi,

The paper is interesting and gives good insight into the world of RFID technology, applications and future possibilities.

Please consider proof reading by a native speaker as the text contains some errors and confusing sentence formulations.

Further on I will go line by line on account of things I found should be corrected:

L18 - either 'in this field' or ' of this field'

L26 - To increase clarity start the introduction with a short definition or explanation of RFID principle

L34 - 'adds increases system cimplexity'

L43 - 'embedding THE sensor'

L68 - 'countRies'

L92 - add ref. to Infineon

L96 - specify what constitutes a 'mean time'. Is there a mean sampling rate included in this mean battery life time.

L126 – some table colors are black?

L131 – figure 4!

L157 – battery x2, ‘they can be set to reach…’ - poor sentence formulations

L158 – and ON its

L169 – what do you mean by controlled? Or do you mean monitored?

L171 – reformulate

L193 – does the energy scavenging in this case recharge or complement the battery

L203 – maybe comment on poor sustainability of this poor concept

L216 – text COPIED from previous page (L184), reformulate!

L234 – Reference!

L277 – Introduce top) bottom) or use more clear description of the caption text c) and d) missing!

L285/286 – ‘measure-and’ Something went wrong!

L339 – ‘signa’, ‘the antenna’ x2 !

L366 – Introduce references in the table

L467 – unit super scripting wrong! cm2 ; -1

L 474 – Check super scripting!

L950 – font size smaller!

Author Response

Hi,

The paper is interesting and gives good insight into the world of RFID technology, applications and future possibilities.

Please consider proof reading by a native speaker as the text contains some errors and confusing sentence formulations.

Thanks, we tried to polish the paper from confusing sentence formulations.

Further on I will go line by line on account of things I found should be corrected:

L18 - either 'in this field' or ' of this field'

Thanks, the sentence has been corrected.

L26 - To increase clarity start the introduction with a short definition or explanation of RFID principle

L34 - 'adds increases system cimplexity'

Thanks, the sentence has been corrected.

L43 - 'embedding THE sensor'

Thanks, the sentence has been corrected.

L68 - 'countRies'

Thanks, the sentence has been corrected.

L92 - add ref. to Infineon

A reference has been added.

L96 - specify what constitutes a 'mean time'. Is there a mean sampling rate included in this mean battery life time.

Mean time is a maintenance metric that measures the average amount of time an asset operates before expiring.

L126 – some table colors are black?

Table colors has been fixed.

L131 – figure 4!

Figure number has been corrected.

L157 – battery x2, ‘they can be set to reach…’ - poor sentence formulations

Thanks, the sentence has been corrected.

L158 – and ON its

Thanks, the sentence has been corrected.

L169 – what do you mean by controlled? Or do you mean monitored?

Thanks, the sentence has been corrected.

L171 – reformulate

Thanks, the sentence has been reformulated.

L193 – does the energy scavenging in this case recharge or complement the battery

The BAP tag is like a normal RFID tag but it also has the opportunity to do other than just identify itself, then with the aid of a battery. Sensor data is collected by the CPU and written to the tag's memory. To access the tag's memory from the CPU the tag need batteries who also prolong the reading distance between the reader and tag. The energy scavenging complements the battery in this case.

L203 – maybe comment on poor sustainability of this poor concept

The sentence has been removed.

L216 – text COPIED from previous page (L184), reformulate!

The sentence has been removed.

L234 – Reference!

A reference has been added.

L277 – Introduce top) bottom) or use more clear description of the caption text c) and d) missing!

Thanks, (c) and (d) has been introduced in the caption.

L285/286 – ‘measure-and’ Something went wrong!

Corrected.

L339 – ‘signa’, ‘the antenna’ x2 !

Corrected.

L366 – Introduce references in the table

A reference has been added.

L467 – unit super scripting wrong! cm2 ; -1

Corrected.

L 474 – Check super scripting!

Corrected.

L950 – font size smaller!

Corrected.

Reviewer 3 Report

The paper is an interesting review on RFID technology and relative IoT applications.

Introduction
I suggest to authors to better organize the section when they describe the RFID system.
the section appears a little bit confusing
In particular, the classification between battery powered RFID nodes should be described and discussed.
the presentation of the different systems should be done, shortly, in a more ordered and schematic way and potential application should be depicted.

row 26 the acronym should be shown at the first occurrence and not in row 59

Section 2 
row 87  The sense is unclear. The sentence should be rephrased 
row 87-91 The sentences should be rephrased and the same concepts referred in a more short way
row 92  the company "Infineon" should be differently cited  for example "(i.e. Infineon, Germany)" 
row 91-92 I guess the "Infineon" is not the only manufacturer of transceiver... the authors should clarify
row 93  the acronym WSN is not described
row 93  why in WSNs the batteries are irreplaceable? The authors should clarify
row 100 US$ does not seem to be a standard definition of US dollars. I suggest "USD"
row 101 replace "a dollar" wit "one dollar"
Table1 the use of different colors tor the text in the table reduces the readability and does not seem improve its
       understandability. High medium and low are written in green, red and black without any apparent specific significance

Section 3
row 184 reading rage?
rows 184-185 the sentence is unclear and should be rephrased. voltage available at the chip  --> value of voltage...
rows 189-190 I suggest to move this sentence, that reports a basic specification of these devices, at the beginning of the section     where their description is given
row 204 The manufacturer of the RFID device should be reported

Other examples of devices with the same features should be reported. Is the Monza x2k the only device on the market with these characteristics? 

rows 216-217 the sentence is unclear and should be rephrased
row 221 voltage available at the chip  --> value of voltage...
row 224 to be the chip... ??  It is unclear and should be rephrased
row 227 the sentence is unclear. In which way the operating range of the ADC can depend on the power consumption rather than the voltage level?
rows 309-310 the sentence is unclear and should be rephrased
row 344 analogously ??
rows 344-350 The authors should give more details about how this kind of tags work. They state that these are like an antenna but nothing more...
Table 2 is split in two pages
row 372 401 418 496 RCS, SAR, FMCW  and FHE acronyms should be explained 
Figure 9 I suggest a more contrasted choice of colors to improve the readability

 Section 6
 row 898 EPC acronym should be explained
 row 898-900 the sentence is unclear and should be rephrased
 row 904-907 are rather hidden and should be moved 
 rows 912-914 and 923-926   the sentence is the same. Fix its
 row 971 the acronym RSSS should be explained

The overall comment is that the paper present a wide collection of RFID technologies- 
The rearrangement of some concepts discussed in the paper in a more ordered scheme could improve its readability and comprehension
The English should be improved especially for some sentences that result unclear.

Author Response

The paper is an interesting review on RFID technology and relative IoT applications.

Introduction
I suggest to authors to better organize the section when they describe the RFID system.
the section appears a little bit confusing
In particular, the classification between battery powered RFID nodes should be described and discussed.
the presentation of the different systems should be done, shortly, in a more ordered and schematic way and potential application should be depicted.

Thanks for the comment. The introduction section has been edited to improve the readability. Battery powered nodes are compared to passive solutions which are the main focus of this review paper.

row 26 the acronym should be shown at the first occurrence and not in row 59

The acronym has been declared at the first occurrence as suggested.

Section 2 
row 87  The sense is unclear. The sentence should be rephrased 

Thanks, the sentence has been reformulated.

row 87-91 The sentences should be rephrased and the same concepts referred in a more short way

Thanks, the sentence has been reformulated.

row 92  the company "Infineon" should be differently cited  for example "(i.e. Infineon, Germany)" 
row 91-92 I guess the "Infineon" is not the only manufacturer of transceiver... the authors should clarify

Thanks, the sentence has been reformulated.

row 93  the acronym WSN is not described

The acronym has been clarified.

row 93  why in WSNs the batteries are irreplaceable? The authors should clarify

In some cases, the WNS nodes have an integrated battery which is not replaceable.

row 100 US$ does not seem to be a standard definition of US dollars. I suggest "USD"

Revised

row 101 replace "a dollar" wit "one dollar"

Revised

Table1 the use of different colors tor the text in the table reduces the readability and does not seem improve its  understandability. High medium and low are written in green, red and black without any apparent specific significance

Revised

Section 3
row 184 reading rage?

Clarified (‘maximum reading distance from the sensor tag’)

rows 184-185 the sentence is unclear and should be rephrased. voltage available at the chip  --> value of voltage...

Clarified (‘… represented by the value of the voltage that is available at the chip’)

rows 189-190 I suggest to move this sentence, that reports a basic specification of these devices, at the beginning of the section     where their description is given

The sentence has been moved before section 3.1.

row 204 The manufacturer of the RFID device should be reported

Name of the manufacturer has been included and as well as a reference to the data sheet.

Other examples of devices with the same features should be reported. Is the Monza x2k the only device on the market with these characteristics? 

Other commercial chips available on the market are reported in Table 7.

rows 216-217 the sentence is unclear and should be rephrased

Reviewed (‘Electronic RFID sensors can exploit HF or UHF frequency range’

row 221 voltage available at the chip  --> value of voltage...

Clarified (‘… represented by the value of the voltage that is available at the chip’)

row 224 to be the chip... ??  It is unclear and should be rephrased

Revised (‘Moreover, the sensor must be able to be miniaturized in order to embedded in the chip.’)

row 227 the sentence is unclear. In which way the operating range of the ADC can depend on the power consumption rather than the voltage level?

ADC has its own power consumption [1] which impacts the UHF RFID tags’ operating range. Even the driving of an ADC with power consumption of several µW will reduce the operating distance.

[1] Sosa, C. (2018). How to use power scaling to maximize power savings in a SAR ADC system. Analog Design Journal, 1-4.

rows 309-310 the sentence is unclear and should be rephrased

Revised (‘The adaptive matching feature exhibited by this kind of RFID tags can also be advantageously exploited for sensing purposes.’)

row 344 analogously ??

Revised (‘Chipless RFID sensor tags, likewise to the electromagnetic RFID sensors, …’)

rows 344-350 The authors should give more details about how this kind of tags work. They state that these are like an antenna but nothing more...

Some additional detail on the working principle of chipless RFID sensors have been added.

Table 2 is split in two pages

Now it is contained in a single page.

row 372 401 418 496 RCS, SAR, FMCW  and FHE acronyms should be explained 

Thanks for the suggestion. The acronyms have been now defined at their first occurrence.

Figure 9 I suggest a more contrasted choice of colors to improve the readability

The background color of figure 9 has been changed to improve the contrast.

 Section 6
 row 898 EPC acronym should be explained

Acronym has been defined: Electronic Product Code (EPC)

 row 898-900 the sentence is unclear and should be rephrased

The sentence has been reformulated.

 row 904-907 are rather hidden and should be moved 

The lines have been moved above the table and reformulated.

rows 912-914 and 923-926   the sentence is the same. Fix its

Thanks, the sentence has been removed in both the occurrences since it was unnecessary.

row 971 the acronym RSSS should be explained

Acronym has been explained: RSSI (Received Signal Strength Indicator).

The overall comment is that the paper present a wide collection of RFID technologies- 
The rearrangement of some concepts discussed in the paper in a more ordered scheme could improve its readability and comprehension

The proposed arrangement of concepts is certainly one of the possible ones. We though it was the most logic one.

The English should be improved especially for some sentences that result unclear.

We tried to improve the English by proofreading the paper once again.

Reviewer 4 Report

This review paper is organized in an interesting way and gives the up to date technologies over the broad range of applications, architectures, strategies and so on.

I found some typos as below. Fix them before final submission.

In line 21: on the market

In line 145: the tag acts like

In line 959: [121, 122],

Author Response

This review paper is organized in an interesting way and gives the up to date technologies over the broad range of applications, architectures, strategies and so on.

Thanks for the positive assessment of the paper.

I found some typos as below. Fix them before final submission.

In line 21: on the market

The typo has been corrected.

In line 145: the tag acts like

The typo has been corrected.

In line 959: [121, 122],

Corrected.

Round 2

Reviewer 3 Report

The paper has been significantly improved in readability and comprehension. Few minor issues should be fixed before the publication

line 33  the acronyms should be declared: HF (NFC) and UHF (RAIN) 
         moreover, in this case, the range of both the frequencies HF and UHF should be indicated 
line 89  Infineon --> Infineon Technologies AG (Germany),    Analog devices --> Analog Devices Inc (USA)
line 185 Impinj --> Impinj Inc (USA)
line 188 DCI pin.  What is the DCI pin?  In my opinion is too specific for the readers not involved in MCU board design - please 
         rephrase in a more readable way. 
line 615 I suggest to add  this reference related to automotive applications 
         Leschke, A., Weinert, F., Obermeier, M., Kubica, S. and Bonaiuto, V., 2020. Method for classification of frontal collision events in passenger cars based on measurement of local component-specific decelerations. International journal of automotive technology, 21(4), pp.785-794.         
line 683 NASA --> US National Space Agency (NASA)
line 711 the acronym GPS even if well known should be declared
line 823 HYGRO-FENIX-RM     the manufacturer should be more clearly reported
line 844 ON Semiconductor and RFMicron -->  ON Semiconductor Inc (USA) and RFMicron, Inc (USA)
Table 7 and 9-10-11  The manufacturers of the sensors should be reported

Author Response

The paper has been significantly improved in readability and comprehension. Few minor issues should be fixed before the publication

Thanks for the positive assessment of the paper.

line 33  the acronyms should be declared: HF (NFC) and UHF (RAIN)

The acronyms have been declared.

         moreover, in this case, the range of both the frequencies HF and UHF should be indicated

The ranges are those licensed for these applications. The bands change from Europe to US. We have reported the central frequencies in the text, but the precise bands are reported in Table 8.

line 89  Infineon --> Infineon Technologies AG (Germany),    Analog devices --> Analog Devices Inc (USA)

line 185 Impinj --> Impinj Inc (USA)

Thanks, we have corrected the names of the companies.

line 188 DCI pin.  What is the DCI pin?  In my opinion is too specific for the readers not involved in MCU board design - please

rephrase in a more readable way.

The DCI PIN is the pin for providing DC voltage. We have rephrased.

line 615 I suggest to add  this reference related to automotive applications

         Leschke, A., Weinert, F., Obermeier, M., Kubica, S. and Bonaiuto, V., 2020. Method for classification of frontal collision events in passenger cars based on measurement of local component-specific decelerations. International journal of automotive technology, 21(4), pp.785-794.        

Reference has been added.

line 683 NASA --> US National Space Agency (NASA)

Corrected.

line 711 the acronym GPS even if well known should be declared

The acronym has been declared.

line 823 HYGRO-FENIX-RM     the manufacturer should be more clearly reported

The product is commercialized by Farsens, Spain. The manufacturer has been reported.

line 844 ON Semiconductor and RFMicron -->  ON Semiconductor Inc (USA) and RFMicron, Inc (USA)

Thanks, we have corrected the names of the companies.

Table 7 and 9-10-11. The manufacturers of the sensors should be reported

In table 7, the manufactures are reported in the first column. In tables 9, 10, 11, they are reported within parentheses in the first column.